# MagCache: Fast Video Generation with Magnitude-Aware Cache

**Zehong Ma**[1,2,†], **Longhui Wei**[2,*,‡], **Feng Wang**[2], **Shiliang Zhang**[1,*], **Qi Tian**[2]

[1] State Key Laboratory of Multimedia Information Processing,
School of Computer Science, Peking University
[2]Huawei Inc.

zehongma@stu.pku.edu.cn, weilh2568@gmail.com,
fwangeve@foxmail.com, slzhang.jdl@pku.edu.cn, tian.qi1@huawei.com

Project page: https://Zehong-Ma.github.io/MagCache

Codes: https://github.com/Zehong-Ma/MagCache

## Abstract

Existing acceleration techniques for video diffusion models often rely on uniform heuristics or time-embedding variants to skip timesteps and reuse cached features. These approaches typically require extensive calibration with curated prompts and risk inconsistent outputs due to prompt-specific overfitting. In this paper, we introduce a novel and robust discovery: a unified magnitude law observed across different models and prompts. Specifically, the magnitude ratio of successive residual outputs decreases monotonically, steadily in most timesteps while rapidly in the last several steps. Leveraging this insight, we introduce a Magnitude-aware Cache (MagCache) that adaptively skips unimportant timesteps using an error modeling mechanism and adaptive caching strategy. Unlike existing methods requiring dozens of curated samples for calibration, MagCache only requires a single sample for calibration. Experimental results show that MagCache achieves 2.10×—2.68× speedups on Open-Sora, CogVideoX, Wan 2.1, and HunyuanVideo, while preserving superior visual fidelity. It significantly outperforms existing methods in LPIPS, SSIM, and PSNR, under similar computational budgets.

## 1 Introduction

In recent years, diffusion models [1, 2, 3, 4] have achieved remarkable success in visual generation and understanding[5, 6] tasks. These models have evolved from U-Net [7, 8, 9] to more sophisticated diffusion transformers [10], significantly enhancing both model capacity and generation quality. Leveraging these advancements, state-of-the-art video generation frameworks [11, 12, 13, 14, 15, 16] have demonstrated impressive fidelity and temporal coherence in generated videos.

Despite these achievements, the slow inference speed of diffusion models remains a critical bottleneck [17]. The primary reason is the inherently sequential nature of the denoising process [18], which becomes increasingly problematic as models scale to higher resolutions and longer video durations [19, 14]. While recent approaches such as distillation [20, 21, 22] and post-training quantization [23, 24] offer potential acceleration, they often require costly retraining and additional data, making them less practical for widespread adoption.

Caching-based methods [25, 26, 27, 28] present a lightweight alternative by reusing intermediate outputs in multiple steps without the need for retraining. However, conventional uniform caching strategies fail to fully exploit the dynamic nature of output similarities during inference, leading

---

*Corresponding authors. ‡ Project leader. † Work was done during internship at Huawei Inc.

39th Conference on Neural Information Processing Systems (NeurIPS 2025).

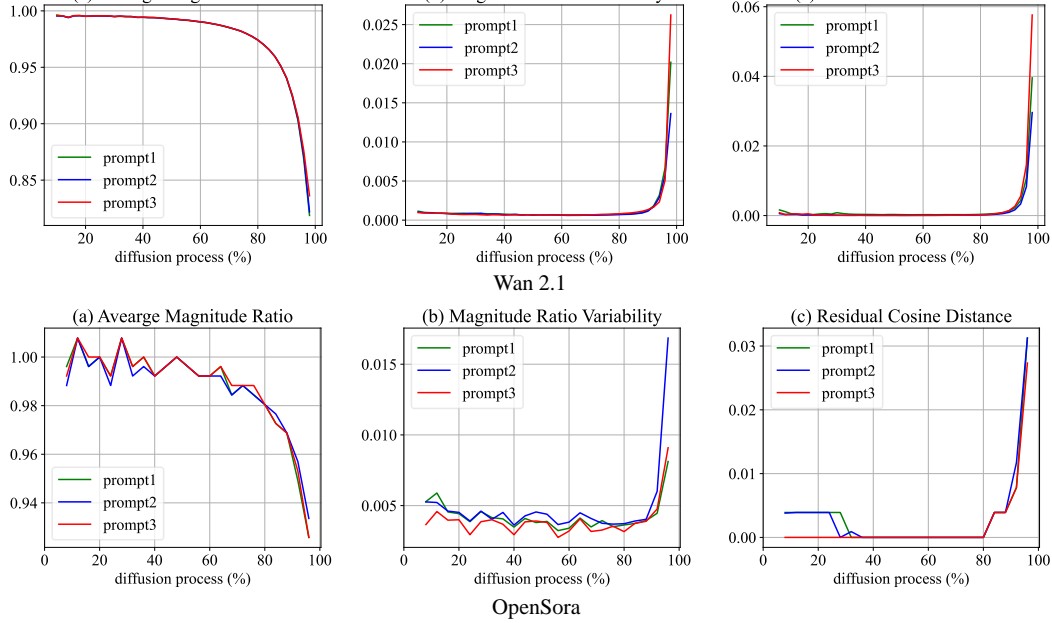

Figure 1: Relationships between residuals across diffusion timesteps. Differences between adjacent residuals are mainly due to magnitude rather than direction during the first 80% of steps. In the final 20%, both magnitude ratio and cosine distance change sharply in opposite trends, but the magnitude ratio still reflects residual differences. (a) Average magnitude ratio decreases gradually, then drops sharply near the end; ratios close to 1 indicate stable transitions suitable for cache reuse. (b) Standard deviation of the magnitude ratio remains near zero in early steps, indicating stable magnitudes. (c) Token-wise cosine distance stays near zero early on, showing consistent residual directions.

to redundant computations and suboptimal cache utilization. Recent work such as AdaCache [29] dynamically adjusts caching strategies based on content complexity, while FasterCache [30] identifies redundancy in classifier-free guidance (CFG) outputs. Additionally, TeaCache [31] builds step-skipping functions through output residual modeling with time embedding difference or modulated input difference, which requires extensive calibration for different models and may overfit to the calibration set.

In this paper, we uncover a new law for the magnitude ratio of successive residual outputs across different video diffusion models and prompts. The residual output is the difference between the model's predicted output and its input. The magnitude ratio shows the change of residuals from previous timestep to the current one. Figure 1(a) shows that the magnitude ratio steadily decreases during most of the diffusion process and drops sharply in the final steps. This trend tells us that many early and mid-range timesteps behave very similarly, which suggests that there is a lot of redundancy that can be used to speed up the process.

Figure 1 demonstrates that the change across adjacent timesteps comes mainly from differences in magnitude rather than from their direction in the first 80% timesteps. In Figure 1(c), the token-wise cosine distances are very small in the first 80% process. This indicates that the direction vectors remain very similar. In addition, Figure 1(b) shows that the standard deviation of the magnitude ratio is close to zero in the early stages. Together, these observations confirm that the change between residual outputs at adjacent timesteps is primarily due to differences in average magnitude in the early timesteps. For the last 20% timesteps, both average magnitude ratio and residual cosine distance change dramatically. It indicates that the residual change of final 20% steps can also be measured by average magnitude ratios. Therefore, we can utilize the unified law of average magnitude ratio to indicate the difference between adjacent timesteps. Besides, the magnitude ratio is robust across various random inputs. As seen in Figure 1(a) for Wan 2.1, the ratio stays consistent and stable when different prompts are used.

By relying on the stable magnitude ratio, we can avoid the complexity and the risk of overfitting found in polynomial fitting methods like TeaCache [31]. With this robust magnitude law, we can more accurately estimate and control the error introduced by skipping timesteps. This allows us to achieve a significant acceleration in inference without compromising visual quality.

Motivated by this insight, we introduce MagCache, a magnitude-aware cache designed to adaptively skip unimportant timesteps based on the observed magnitude ratios. The method consists of two main components:

*Accurate Error Modeling*: Building on the observations in Figure 1, we model the potential error introduced by skipping timesteps by quantifying changes in residual magnitude. Unlike TeaCache [31], our method accurately estimates the error even when multiple consecutive steps are skipped. In contrast, TeaCache performs poorly in this scenario due to the inherent inaccuracies of its polynomial fitting and prediction approach. By leveraging magnitude variation to estimate error, MagCache ensures that timestep skipping does not significantly compromise the quality of the generated video.

*Adaptive Caching Strategy*: With the accurate error modeling, we can adaptively skips consecutive timesteps until its accumulated error exceeds the predefined threshold or maximum skip length. This ensures that the total approximation error remains within an acceptable threshold, maintaining high visual fidelity while achieving significant speedup.

In contrast to the TeaCache, which using 70 curated prompts to fitting coefficients, our MagCache requires only a random sample to forward once for calibration, avoids extensive fitting time. Besides, the calibrated magnitude curve is more stable and robust than the polynomial curve. It seamlessly integrates into existing diffusion model pipelines, providing a plug-and-play solution for efficient video generation. Our contributions are summarized as follows:

- **Unified Magnitude Law**: We identify a stable, monotonically decreasing ratio of residual magnitudes, which is robust across different prompts, providing a principled criterion for skipping redundant diffusion steps during inference.

- **MagCache**: We introduce MagCache, a magnitude-aware cache that adaptively skips timesteps with an error modeling mechanism and adaptive caching strategy.

- **Superior Performance**: MagCache consistently achieves over $2\times$ inference speedup on video diffusion models such as Open-Sora, CogVideoX, Wan 2.1, and HunyuanVideo, as well as on image diffusion model Flux, while maintaining superior visual fidelity. Under similar computational budgets, MagCache significantly outperforms existing caching-based methods across LPIPS, SSIM, and PSNR metrics.

## 2 Related Work

### 2.1 Diffusion Models for Video Synthesis

Diffusion models have become foundational in generative modeling due to their ability to produce high-quality and diverse outputs [2, 3, 32, 33]. Initially, diffusion models for video synthesis followed the U-Net-based architecture and extended image diffusion models for temporally coherent video generation [34, 7, 35, 36, 37, 38, 39]. These methods generated short to medium-length videos by conditioning spatial diffusion models on temporal signals [40, 41].

However, the scalability of U-Net-based architectures poses limitations in modeling complex spatiotemporal dependencies. Inspired from the success of LLM [42, 43, 44, 45, 46, 47], transformer-based diffusion models (DiT)[10] have been increasingly adopted due to their greater modeling capacity and flexibility[48, 19, 13, 49, 14]. Notably, Open-Sora [50] demonstrates the scalability and realism achievable through large-scale training of diffusion transformers for video generation. Recently, the newly open-sourced Wan 2.1 [16] has demonstrated impressive video generation performance, but generating a five-second video still takes several minutes on a single A800 GPU.

### 2.2 Efficiency Improvements in Diffusion Models

Despite their impressive generation quality, diffusion models suffer from high inference costs, which limit their deployment in real-time or resource-constrained settings. Efforts to improve efficiency can

be broadly categorized into reducing the number of sampling steps and lowering the computational cost per step.

For sampling step reduction, methods based on improved SDE or ODE solvers [51, 52, 53, 54] and model distillation [55, 22, 20, 21] have been proposed. Consistency models [56, 57] and pseudo-numerical solvers [4, 58] offer further improvements for fast sampling. Caching-based methods improve inference efficiency by reusing features at select timesteps.

To reduce per-step cost, approaches such as quantization [59, 60, 61, 62], pruning [63, 64], sparse attention[65, 66, 67], and neural architecture search [68, 69] have been explored.

In these methods, cache-based acceleration [70, 71, 72] has gained attention due to its simplicity and portability. DeepCache [25], Faster Diffusion [73], and PAB [27] improve inference efficiency by reusing features at select timesteps. $\Delta$-DiT [28] adapts this idea to transformer-based models by caching attention-layer residuals. Recent work such as AdaCache [29] dynamically adjusts caching strategies based on content complexity, while FasterCache [30] identifies redundancy in classifier-free guidance (CFG) outputs to enable efficient reuse.

While these cache-based approaches demonstrate promising results, they often rely on heuristic or data-driven patterns that may not generalize across prompts or model variants. For example, TeaCache [31] builds step skipping functions through prompt-specific residual modeling with 70 curated prompts. It may overfit the calibration prompts and require extensive resources for calibration.

**Differences with Previous Methods**: Our method leverages a newly discovered unified law in residual magnitudes to accurately control the error when skipping timesteps. Unlike TeaCache, which requires extensive prompt-specific polynomial fitting and calibration, our approach only needs a single random sample for calibration. This simpler, magnitude-aware strategy offers a more robust and generalizable caching mechanism. It achieves significant speedup without compromising visual fidelity, and it reliably performs across different models and scenarios.

## 3  Method

### 3.1  Preliminary

**Flow Matching.** Flow matching [74, 75] is a continuous-time generative modeling framework that learns a velocity field $\mathbf{v}_\theta(\mathbf{x}, t)$ to transport samples from a data distribution $p_{\text{data}}(\mathbf{x}_0)$ to a simple prior distribution (e.g., Gaussian). Given a forward trajectory $\mathbf{x}_t$ defined by a stochastic or deterministic interpolation between $\mathbf{x}_0$ and a noise sample $\mathbf{x}_1$, the training objective is to match the model-predicted velocity $\mathbf{v}_\theta(\mathbf{x}_t, t)$ to the target velocity $\mathbf{v}^*(\mathbf{x}_t, t)$:

$$\mathcal{L}_{\text{FM}} = \mathbb{E}_{\mathbf{x}_0, \mathbf{x}_1, t} \left[ \|\mathbf{v}_\theta(\mathbf{x}_t, t) - \mathbf{v}^*(\mathbf{x}_t, t)\|^2 \right]. \tag{1}$$

This formulation naturally encompasses diffusion models and score-based models as special cases. For example, linear interpolants with time-dependent velocity targets can recover DDPM and score matching objectives.

**Trajectory and Velocity.** The forward trajectory $\mathbf{x}_t$ is constructed via a prescribed interpolant between the data sample $\mathbf{x}_0$ and a noise sample $\mathbf{x}_1 \sim \mathcal{N}(0, \mathbf{I})$, such as:

$$\mathbf{x}_t = (1 - \rho(t))\, \mathbf{x}_0 + \rho(t)\, \mathbf{x}_1, \tag{2}$$

where $\rho(t)$ is a monotonically increasing interpolation schedule with $\rho(0) = 0$ and $\rho(1) = 1$. The ground-truth velocity $\mathbf{v}^*(\mathbf{x}_t, t)$ can be derived from the time derivative of $\mathbf{x}_t$ or computed via an optimal transport formulation depending on the specific design.

**Residual.** In this work, we define the *residual* as the difference between the model's predicted velocity and its input at each timestep:

$$\mathbf{r}_t = \mathbf{v}_\theta(\mathbf{x}_t, t) - \mathbf{x}_t. \tag{3}$$

This residual captures the effective "update signal" generated by the model at each step, which reflects the model's internal belief about how the input should evolve. By analyzing these residuals across different timesteps, we uncover the magnitude correlation with timesteps, which form the basis for our MagCache introduced in Section 3.3.

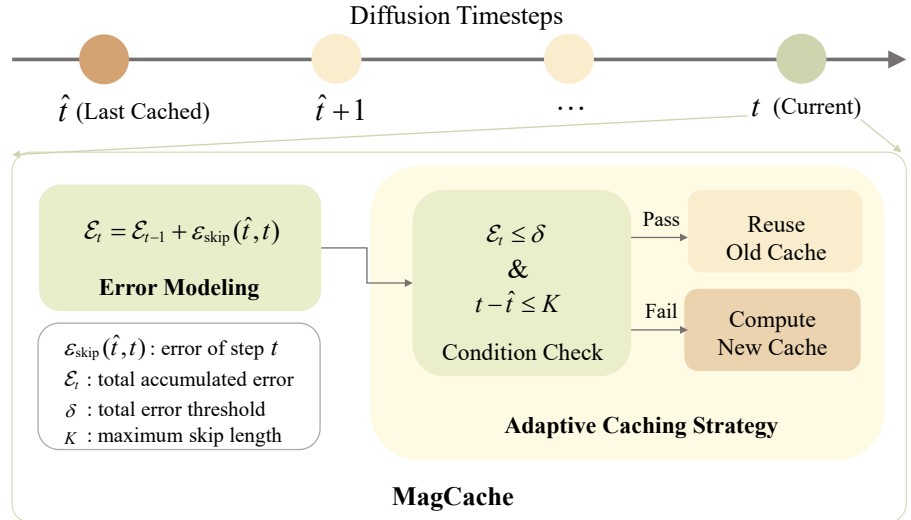

Figure 2: Overview of the MagCache. The MagCache consists of error modeling mechanism and adaptive caching strategy. With the estimated total accumulated error $\mathcal{E}$, MagCache adaptively reuses the old cache or computes a new cache by validating the two conditions in Sec 3.3.

## 3.2 Magnitude Analysis

In this section, we empirically demonstrate that our unified magnitude law serves as both an accurate and stable criterion for measuring the difference between residuals. Concretely, we show (1) that the average magnitude ratio faithfully captures the change in residual outputs, and (2) that this ratio exhibits remarkable consistency across different models and prompts.

We define the per-step magnitude ratio as

$$\gamma_t = \text{mean}\left(\frac{\|\mathbf{r}_t\|_2}{\|\mathbf{r}_{t-1}\|_2}\right), \tag{4}$$

where $\mathbf{r}_t = \mathbf{v}_\theta(\mathbf{x}_t, t) - \mathbf{x}_t$ denotes the residual at timestep $t$.

**Accurate Criterion for Residual Difference.** First, we verify that changes in residual outputs between adjacent timesteps are driven almost entirely by differences in magnitude rather than direction. As shown in Figure 1(a), during the first 80% of the diffusion trajectory, $\gamma_t$ decreases slowly and smoothly from 1, and the standard deviation of $\gamma_t$ remains near zero (Figure 1(b)). Meanwhile, the token-wise cosine distance between $\mathbf{r}_t$ and $\mathbf{r}_{t-1}$ also stays extremely small (Figure 1(c)), indicating that their directional patterns are virtually unchanged. Together, these observations confirm that

$$\|\mathbf{r}_t - \mathbf{r}_{t-1}\| \approx \big|\|\mathbf{r}_t\| - \|\mathbf{r}_{t-1}\|\big|, \tag{5}$$

so that $\gamma_t$ alone accurately quantifies the residual difference. In practice, when $\gamma_t$ is close to 1, the two residuals are nearly identical; when $\gamma_t$ drops, their difference grows accordingly.

**Robustness Across Models and Prompts.** Next, we assess the robustness of $\gamma_t$ under varying models and textual prompts. Figure 1(a) overlays the average magnitude-ratio curves for both Wan 2.1 and Open-Sora models: in each case, $\gamma_t$ follows the same monotonically decreasing trajectory, with only a sharp fall in the final few steps. This consistency demonstrates model-agnostic behaviour. We further sample a diverse set of text prompts (see Appendix) and compute $\gamma_t$ for each; the resulting curves almost coincide on Wan 2.1, showing that $\gamma_t$ is prompt-invariant. Such stability implies that a single calibration run suffices to characterize residual scaling for any prompt or model variant.

By combining these two findings, we establish that the average magnitude ratio not only captures the true per-step change in residual outputs but does so in a highly stable and prompt-agnostic manner. This insight underpins our adaptive caching mechanism in Section 3.3, allowing MagCache to skip redundant timesteps reliably without risking undue approximation error.

### 3.3 MagCache

Building on the unified magnitude law analyzed in Section 3.2, MagCache leverages the stable, monotonically decreasing behavior of the per-step magnitude ratio $\gamma_t$ to drive both its error modeling and adaptive caching decisions. An overview is shown in Figure 2.

#### 3.3.1 Error Modeling

Let $\hat{t}$ be the last timestep at which we refreshed the cache. If we skip steps $\hat{t} + 1, \dots, t$, the skip error at step $t$ is given by

$$\varepsilon_{\text{skip}}(\hat{t}, t) = 1 - \prod_{i=\hat{t}+1}^{t} \gamma_i, \tag{6}$$

where

$$\gamma_i = mean(\frac{\|\mathbf{r}_i\|_2}{\|\mathbf{r}_{i-1}\|_2}). \tag{7}$$

Since Section 3.2 demonstrated that (i) the residual difference is dominated by these magnitude changes (Equation 5 ), and (ii) $\gamma_i$ is highly stable (low variance) across models and prompts, this multiplicative estimate closely matches the practical deviation between the cached residual $\mathbf{r}_{\hat{t}}$ and the ground-truth residual $\mathbf{r}_t$. To account for accumulated error over multiple skips, we maintain a running total error $\mathcal{E}_t$:

$$\mathcal{E}_t = \mathcal{E}_{t-1} + \varepsilon_{\text{skip}}(\hat{t}, t), \tag{8}$$

initialized with $\mathcal{E}_{\hat{t}} = 0$. Thanks to the near-zero standard deviation of $\gamma_i$ in early and mid timesteps (Figure 1(b)), this accumulation remains predictable, ensuring our bound on approximation error is both tight and reliable.

#### 3.3.2 Adaptive Caching Strategy

Armed with an accurate error estimate, MagCache only skips a step $t$ if both the total accumulated error and the number of consecutive skips remain within bounds. Specifically, it requires

$$\mathcal{E}_t \leq \delta, \tag{9}$$

where $\delta$ is a user-specified threshold on total accumulated error, and

$$t - \hat{t} \leq K, \tag{10}$$

where $K$ is the maximum number of steps that could be skipped using a single cached residual. If either condition is violated, we reset:

$$\hat{t} \leftarrow t, \quad \mathcal{E}_t \leftarrow 0, \tag{11}$$

recompute the true residual $\mathbf{r}_t$, and update the cache. Otherwise, we reuse $\mathbf{r}_{\hat{t}}$ for step $t$, incurring no new computation.

The introduction of a maximum skip length $K$ is crucial. Although our magnitude-based error modeling is highly accurate, it is still an approximation. Over long sequences of steps, small modeling errors can accumulate. By bounding the skip length, we ensure that such drift is regularly corrected, preventing the model from deviating too far from the true residual trajectory.

In summary, by tightly integrating the empirical magnitude law from Section 3.2 into both error modeling and adaptive caching strategy, MagCache provides a principled, training-free acceleration framework that dynamically balances efficiency and quality in video diffusion inference.

## 4 Experiment

### 4.1 Settings

**Base Models and Compared Methods.** To demonstrate the effectiveness of our method, we quantitatively evaluate our MagCache on video diffusion models like Open-Sora 1.2 [11], CogVideoX [14], Wan 2.1 [16], and HunyuanVideo [76] and image diffusion model Flux [77]. Following TeaCache [31],

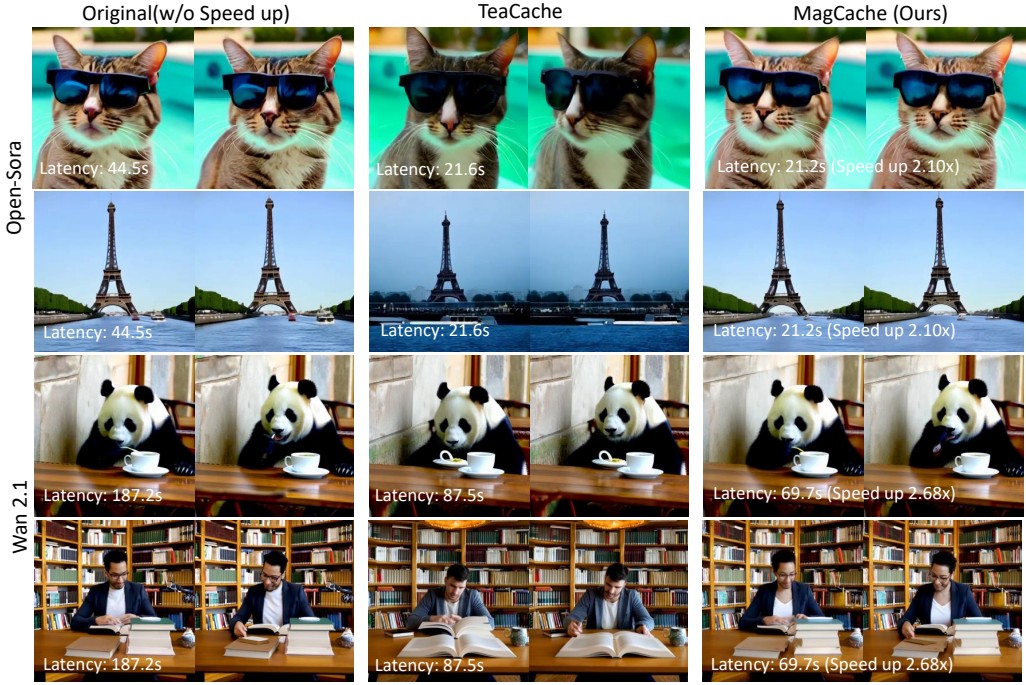

Figure 3: Comparison of visual quality and efficiency (denoted by latency) with the competing method. MagCache outperforms TeaCache [31] in both visual quality and efficiency. Latency is evaluated on a single A800 GPU. Video generation specifications: Open-Sora [11] (51 frames, 480p), Wan 2.1 1.3B [16] (81 frames , 480p). Best-viewed with zoom-in.

we compare our base models with recent efficient video synthesis techniques, including PAB [27], T-GATE [72], $\Delta$-DiT [28], FasterCache [30], TeaCache [31], DuCa [78], and TaylorSeer [79], to highlight the advantages of our approach. Notably, our MagCache also support recent visual generation or editing models, such as FramePack [80], Wan2.2 [16], Flux-Kontext [81], OmniGen2 [82], Qwen-Image [83], and Qwen-Image-Edit [83] in the official code repository.

**Evaluation Metrics.** To assess the performance of video synthesis acceleration methods, we focus on two primary aspects: inference efficiency and visual quality. For evaluating inference efficiency, we use Floating Point Operations (FLOPs) and inference latency as metrics. Following PAB [27] and TeaCache [31], we employ LPIPS [84], PSNR, and SSIM for visual quality evaluation.

**Implementation Detail.** We enable FlashAttention [85] by default for all experiments. Latency is measured on a single A800 GPU. As shown in Figure 1, the magnitude ratio remains stable and robust across different prompts. Therefore, we select the prompt 1 in Appendix A.1 to compute the magnitude ratio. For all models, following prior works [86, 87], we keep the first 20% of diffusion steps unchanged, as these initial steps are critical to the overall generation process. It is consistent with our observation that the magnitude ratio has a relatively larger variation in the first 20% steps. For Open-Sora, we set $K = 3$ and $\delta = 0.12$ for MagCache-fast, and $K = 1$, $\delta = 0.06$ for MagCache-slow. For Wan 2.1, MagCache-fast uses $K = 4$ and $\delta = 0.12$, while MagCache-slow uses $K = 2$ and $\delta = 0.12$. In the ablation study, we randomly sample 100 prompts from VBench to conduct our experiments. More details can refer to our official repository.

## 4.2 Main Results

**Quantitative Comparison.** Table 1 provides a comprehensive evaluation of our proposed MagCache method, highlighting its superiority over TeaCache and other cache-based methods in both inference efficiency and visual quality across diverse scenarios. We evaluated both the slow and fast variants of MagCache on multiple baselines, including Open-Sora 1.2 (51 frames, 480P), Wan 2.1 1.3B (81

Table 1: Quantitative evaluation of inference efficiency and visual quality in video generation models. MagCache consistently achieves superior efficiency and better visual quality across different base models. It surpasses existing methods in visual quality by a large margin under the similar computation budget. † denotes that these methods are not memory-efficient, which yeild tens of additional memory cost.

| Method | Efficiency | | | Visual Quality | | |
|---|---|---|---|---|---|---|
| | FLOPs (P) ↓ | Speedup ↑ | Latency (s) ↓ | LPIPS ↓ | SSIM ↑ | PSNR ↑ |
| **Open-Sora 1.2** (51 frames, 480P) | | | | | | |
| Open-Sora 1.2 ($T=30$) | 3.15 | 1× | 44.56 | - | - | - |
| Δ-DiT [28] | 3.09 | 1.03× | - | 0.5692 | 0.4811 | 11.91 |
| T-GATE [72] | 2.75 | 1.19× | - | 0.3495 | 0.6760 | 15.50 |
| PAB-slow [27] | 2.55 | 1.33× | 33.40 | 0.1471 | 0.8405 | 24.50 |
| PAB-fast [27] | 2.50 | 1.40× | 31.85 | 0.1743 | 0.8220 | 23.58 |
| FasterCache † [30] | 1.91 | 1.72× | 25.90 | 0.1511 | 0.8255 | 23.23 |
| DuCa † [78] | - | 2.08× | 21.42 | 0.2316 | 0.7652 | 19.96 |
| TeaCache-slow [31] | 2.40 | 1.40× | 31.69 | 0.1303 | 0.8405 | 23.67 |
| TeaCache-fast [31] | 1.64 | 2.05× | 21.67 | 0.2527 | 0.7435 | 18.98 |
| MagCache-slow | 2.40 | 1.41× | 31.48 | **0.0827** | **0.8859** | **26.93** |
| MagCache-fast | **1.64** | **2.10×** | **21.21** | 0.1522 | 0.8266 | 23.37 |
| **Wan 2.1 1.3B** (81 frames, 480P) | | | | | | |
| Wan 2.1 ($T=50$) | 8.21 | 1× | 187.21 | - | - | - |
| TeaCache-slow [31] | 5.25 | 1.59× | 117.20 | 0.1258 | 0.8033 | 23.35 |
| TeaCache-fast [31] | 3.94 | 2.14× | 87.55 | 0.2412 | 0.6571 | 18.14 |
| TaylorSeer(N=2, O=1) † [79] | - | 2.07× | 90.15 | 0.3792 | 0.5220 | 15.06 |
| MagCache-slow | 3.94 | 2.14× | 87.27 | **0.1206** | **0.8133** | **23.42** |
| MagCache-fast | **3.11** | **2.68×** | 69.75 | 0.1748 | 0.7490 | 21.54 |
| **HunyuanVideo** (129 frames, 540P) | | | | | | |
| HunyuanVideo ($T=50$) | 45.93 | 1× | 1163 | - | - | - |
| TeaCache-slow [31] | 27.56 | 1.63× | 712 | 0.1832 | 0.7876 | 23.87 |
| TeaCache-fast [31] | 20.21 | 2.26× | 514 | 0.1971 | 0.7744 | 23.38 |
| MagCache-slow | 20.21 | 2.25× | 516 | **0.0377** | **0.9459** | **34.51** |
| MagCache-fast | **18.37** | **2.63×** | **441** | 0.0626 | 0.9206 | 31.77 |
| **CogVideoX 2B** (49 frames, 480P) | | | | | | |
| CogVideoX ($T=50$) | 2.36 | 1× | 74.10 | - | - | - |
| TeaCache [31] | 1.03 | 2.30× | 32.20 | 0.1221 | 0.8815 | 27.08 |
| MagCache | 0.99 | **2.37×** | **31.15** | **0.0787** | **0.9210** | **30.44** |
| **Flux** (Text-to-Image 1024 × 1024) | | | | | | |
| Flux ($T=28$) | 1.66 | 1× | 14.26 | - | - | - |
| TeaCache-slow [31] | 0.77 | 2.00× | 7.11 | 0.2687 | 0.7746 | 20.14 |
| TeaCache-fast [31] | 0.59 | 2.52× | 5.65 | 0.3456 | 0.7021 | 18.17 |
| MagCache-slow | 0.59 | 2.57× | 5.53 | **0.2043** | **0.8883** | **24.46** |
| MagCache-fast | **0.53** | **2.82×** | **5.05** | 0.2635 | 0.8093 | 21.35 |

frames, 480P), Flux (Text-to-Image 1024×1024) and HunyuanVideo (129 frames, 540p) to provide a robust comparison.

On the Open-Sora 1.2 benchmark, compared to TeaCache-slow, with an LPIPS of 0.1303, MagCache-slow significantly improves visual quality with an LPIPS of 0.0827, an SSIM of 0.8859, and a PSNR of 26.93—demonstrating a clear advantage over TeaCache. Notably, our MagCache-fast variant achieves a remarkable 2.10× speedup with a latency of 21.21 seconds. This performance is comparable to the TeaCache-slow variant, which operates with a higher latency of 31.69 seconds, while both methods deliver similar visual quality. In other words, MagCache-fast successfully combines a high acceleration effect with performance that rivals TeaCache-slow, demonstrating that it is possible to achieve both rapid inference and competitive visual fidelity simultaneously.

For the Wan 2.1 1.3B benchmark, the benefits of MagCache become even more apparent. MagCache-slow reduces FLOPs from 8.21 to 3.94, resulting in a 2.14× speedup and a latency of 87.27 seconds, compared to TeaCache-slow's 5.25 FLOPs with a 1.59× speedup and 117.20s latency. In addition, MagCache-slow achieves better visual quality (LPIPS 0.1206, SSIM 0.8133, PSNR 23.42) than its TeaCache counterpart. Meanwhile, MagCache-fast further improves performance by reducing

FLOPs to 3.11, leading to an impressive 2.68× speedup and latency as low as 69.75 seconds, clearly outperforming TeaCache-fast in the speed-accuracy trade-off.

Across other benchmarks, MagCache consistently provides better visual quality under comparable computational budgets. It is worth noting that other methods, such as FasterCache [30], DuCa [78], and TaylorSeer [79], require significantly larger memory for caching, which limits their applicability to video diffusion models. For instance, TaylorSeer requires 40 GB of additional memory to generate a 480P video with Wan 2.1 1.3B, whereas MagCache requires only 0.5 GB of extra memory.

Overall, these results demonstrate that MagCache can achieve better visual quality than other cache-based methods under similar computational cost.

**Compatibility with other acceleration methods.** As shown in Appendix A.3, MagCache is compatible with other acceleration techniques, including model distillation and low-precision arithmetic.

**Visualization.** Figure 3 compares videos generated by MagCache-fast, the original model, and TeaCache-fast[31]. For Open-Sora, TeaCache performs poorly—the overall color and style of the video shift significantly. As shown in Table 1, TeaCache-fast yields very low PSNR scores, indicating poor video quality. When PSNR falls below 20, visual distortions typically become quite noticeable. In the case of Wan 2.1, TeaCache alters key details such as the object held by the panda and the background wall, whereas our method preserves these fine details effectively. In human-centric scenarios, our approach maintains the identity and structure of the person, while TeaCache often modifies the person's identity entirely. Finally, our method achieves a 2.68× speedup on Wan 2.1 without noticeable quality degradation. These results demonstrate that MagCache delivers superior visual quality with reduced latency compared to TeaCache. Please refer to Appendix A.6 for more visualization.

## 4.3 Ablation Studies

Table 2: Ablation study of maximum skip length $K$ and total error threshold $\delta$ in slow and fast inference mode. $K$ controls the acceleration mode, while $\delta$ fine-tunes the qulity-speed trade-off within that mode.

| Mode | $K$ | $\delta$ | Speedup ↑ | LPIPS ↓ | SSIM ↑ | PSNR ↑ |
|---|---|---|---|---|---|---|
| MagCache-slow (Wan2.1 1.3B) | 2 | 0.06 | 2.0× | 0.0940 | 0.8383 | 24.57 |
| | | 0.12 | 2.1× | 0.1053 | 0.8275 | 24.32 |
| | | 0.03 | 1.9× | 0.0888 | 0.8427 | 24.68 |
| MagCache-fast (Wan2.1 1.3B) | 4 | 0.06 | 2.4× | 0.1375 | 0.7749 | 22.34 |
| | | 0.12 | 2.7× | 0.1625 | 0.7571 | 22.25 |
| | | 0.03 | 2.0× | 0.1263 | 0.7828 | 22.51 |
| MagCache-slow (OpenSora) | 2 | 0.06 | 1.8× | 0.1414 | 0.8142 | 23.91 |
| | | 0.12 | 1.9× | 0.1432 | 0.8130 | 23.81 |
| | | 0.03 | 1.7× | 0.1389 | 0.8162 | 24.05 |
| MagCache-fast (OpenSora) | 4 | 0.06 | 2.1× | 0.2040 | 0.7632 | 21.93 |
| | | 0.12 | 2.4× | 0.2065 | 0.7542 | 21.77 |
| | | 0.03 | 2.0× | 0.2000 | 0.7668 | 22.00 |

We conduct an ablation study to analyze the sensitivity of MagCache to its two primary hyperparameters: the maximum skip length $K$ and the total accumulated error threshold $\delta$. The parameter $K$ primarily controls the acceleration mode (i.e., slow mode or fast mode), while $\delta$ serves to fine-tune the trade-off between generation quality and inference speed within that mode. The results in Table 2 demonstrate that a desired speedup can be achieved with only a few adjustments. We provide robust default parameters for a slow mode ($K = 2, \delta = 0.06$) and a fast mode ($K = 4, \delta = 0.06$). Note that in the implementation details of Section 4.1, the parameters were specifically set to achieve similar acceleration speeds to TeaCache [31] for a fair comparison.

**The impact of maximum skip length $K$.** The parameter $K$ primarily governs the acceleration mode of MagCache, setting a broad trade-off between inference speed and generation quality. As shown in Table 2, switching from $K = 2$ to $K = 4$ consistently yields a significant increase in speedup across different models. For instance, with the Wan2.1 model, increasing $K$ from 2 to 4 elevates the speedup

from $2.0\times$ to $2.4\times$ under the default $\delta = 0.06$. This is accompanied by a predictable trade-off in quality metrics, with LPIPS increasing from 0.0940 to 0.1375. This clear relationship allows users to first select an acceleration range by choosing $K$ before detailedly adjusting the qulity-speed trade-off.

**The impact of threshold $\delta$.** Within a selected mode, the threshold $\delta$ offers fine-grained control over the quality-speed trade-off. For example, in the slow mode ($K = 2$) for OpenSora, decreasing $\delta$ from default 0.06 to 0.03 slightly lowers the speedup from $1.8\times$ to $1.7\times$ but improves generation quality, with LPIPS decreasing from 0.1414 to 0.1389 and PSNR increasing from 23.91 to 24.05. Conversely, increasing $\delta$ from 0.06 to 0.12 boosts the speedup to $1.9\times$ at a minor cost to quality. This consistent and monotonic behavior confirms that only 1–2 adjustments to $\delta$ are typically sufficient to achieve a desired balance. This tuning process is efficient and user-friendly, particularly with our interactive ComfyUI integration, where users can observe the effects of parameter changes in minutes.

**The influence of the calibration prompt.** In Table 3, we compare three calibration strategies: calibrate with a random prompt (Prompt 1 from Figure 1), calibrate with all 944 prompts (average magnitude ratios), calibrate with the most distant outlier prompt (farthest from the average curve). The results in Table 3 show that all three configurations yield nearly identical visual quality and speedup, with negligible differences in LPIPS, SSIM, and PSNR. These findings confirm that MagCache does not rely on carefully selected calibration prompts, and is robust even in the presence of outliers or complex inputs.

Table 3: The influence of the calibration prompt. The calibration of MagCache is robust to the random prompt, even to outliers.

| Calibration Prompt | Speedup ↑ | LPIPS ↓ | SSIM ↑ | PSNR ↑ |
|---|---|---|---|---|
| Random Prompt 1 (Ours) | 2.14× | 0.1206 | 0.8133 | 23.42 |
| 944 Prompts | 2.14× | **0.1162** | **0.8163** | **23.52** |
| Outlier Prompt | 2.21× | 0.1209 | 0.8103 | 23.36 |

Please refer to Appendix A.4 for more ablation studies.

# 5 Conclusion and Future Work

In this paper, we introduce MagCache, a novel magnitude-aware cache designed to accelerate video diffusion models by adaptively skipping unimportant timesteps. Our approach leverages a newly discovered unified law governing the magnitude ratio of successive residual outputs, which remains robust across different video samples and prompts. This insight allows us to model skipping errors accurately, ensuring high visual fidelity even during rapid inference. Through extensive evaluations on benchmarks such as Open-Sora and Wan 2.1, we demonstrated that MagCache consistently achieves significant speedups while improving visual quality compared to existing methods. Our results indicate that MagCache is a versatile solution, effectively balancing computation efficiency with output quality, making it applicable in various real-time or resource-constrained video generation scenarios. We have only verified the effectiveness of the magnitude law and MagCache on video generation models. It is necessary to further validate and extend them to more tasks and models. In future work, we will validate the MagCache on more tasks and models.

# Acknowledgements

This work is supported in part by Grant No. 2023-JCJQ-LA-001-088, in part by the Natural Science Foundation of China under Grant No. U20B2052, 61936011, 62236006, in part by the Okawa Foundation Research Award, in part by the Ant Group Research Fund, and in part by the Kunpeng&Ascend Center of Excellence, Peking University.

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

# A Technical Appendices and Supplementary Material

## A.1 Prompts in Figure 1

We utilize the following three prompts to generate the average magnitude ratio, magnitude ratio variability, and residual cosine distance. In all experiments, we only utilize *Prompt 1* to calibrate the average magnitude ratio.

---

- *Prompt 1*: A stylish woman walks down a Tokyo street filled with warm glowing neon and animated city signage. She wears a black leather jacket, a long red dress, and black boots, and carries a black purse. She wears sunglasses and red lipstick. She walks confidently and casually. The street is damp and reflective, creating a mirror effect of the colorful lights. Many pedestrians walk about.
- *Prompt 2*: In a still frame, a stop sign
- *Prompt 3*: a laptop, frozen in time

---

Table 4: The list of prompts in Figure 1. In all experiments, we only utilize prompt 1 to calibrate the magnitude ratio for MagCache.

## A.2 Definition of Statistics in Figure 1

In Figure 1, we define three metrics: the average magnitude ratio, magnitude ratio variability, and residual cosine distance. The average magnitude ratio $\gamma$ is defined in Equation 7. Specifically, Equation 7 first computes the L2 norm of the residuals $\mathbf{r}_t$ and $\mathbf{r}_{t-1}$ along the channel dimension, then takes the token-wise ratio, and finally averages the result across the sequence length dimension to obtain $\gamma_t$. *The mean operation is omitted in Equation 7.*

The magnitude ratio variability $\sigma$ and residual cosine distance $dist$ can be represented as follows:

**Magnitude Ratio Variability.**

$$\sigma_t = std(\frac{\|\mathbf{r}_t\|_2}{\|\mathbf{r}_{t-1}\|_2}),$$

(12)

where $\mathbf{r}_t \in \mathbb{R}^{N \times d}$ denotes the residual at timestep $t$, and $\|\cdot\|_2$ represents the L2 norm computed along the channel dimension $d$. The standard deviation is then calculated across the sequence length dimension $N$.

**Residual Cosine Distance.**

$$dist_t = \frac{1}{N} \sum_{i}^{N} (1 - cos(\mathbf{r}_t^i, \mathbf{r}_{t-1}^i)).$$

(13)

Here, the cosine distance is computed for each token between residuals at timesteps $t$ and $t - 1$, and the final residual cosine distance $dist_t$ is obtained by averaging across all tokens.

## A.3 Compatibility with Other Acceleration Methods

To assess the versatility of MagCache, we investigate its compatibility with other acceleration techniques for diffusion models, namely model distillation and low-precision arithmetic. These experiments demonstrate that MagCache can be seamlessly integrated as a plug-and-play module to achieve further speedups on already optimized models.

**Compatibility with Model Distillation.** Model distillation reduces the number of denoising steps by distillation post-training. We evaluated the performance of MagCache on FusionX, a few-step distilled variant of the Wan2.1 14B model. As presented in Table 5, MagCache achieves a 1.66× speedup on FusionX without a significant degradation in visual quality.

With the reduced step-wise redundancy inherent in distilled models, MagCache still provides substantial acceleration by caching and reusing intermediate computations. Notably, a simple reduction of inference steps to $T = 6$ to match MagCache's speedup results in a severe decline in generation

quality, with an LPIPS score of 0.2982. In contrast, MagCache achieves a much better LPIPS of 0.1812, demonstrating its ability to maintain quality while accelerating inference.

Table 5: Evaluation of MagCache on the few-step distilled model Wan2.1 14B FusionX (33 frames, 480P). MagCache accelerates the distilled model while maintaining significantly better visual quality compared to a baseline with a reduced step count at the same speedup.

| Method | Speedup ↑ | Latency (s) ↓ | LPIPS ↓ | SSIM ↑ | PSNR ↑ |
|---|---|---|---|---|---|
| FusionX ($T$=10) | 1× | 30 | - | - | - |
| FusionX ($T$=6) | 1.66× | 18 | 0.2982 | 0.6471 | 20.35 |
| MagCache ($T$=10, skip 4 steps) | **1.66×** | **18** | **0.1812** | **0.7868** | **24.23** |

**Compatibility with Low-Precision Arithmetic.** We also assessed the performance of MagCache in low-precision settings by applying it to a 4-bit quantized version of the Wan2.1 14B model. The results, detailed in Table 6, show that MagCache remains effective under quantization. Both the fast and slow variants of MagCache improve the quality-speed trade-off.

Specifically, MagCache-fast achieves a 2× speedup with only a minor increase in memory footprint. While there is a slight drop in the speedup ratio compared to the full-precision model due to the reduced numerical precision, the acceleration benefit remains substantial.

Table 6: Evaluation of MagCache on the 4-bit quantized Wan2.1 14B model (33 frames, 480P). MagCache is compatible with low-precision settings, maintaining its acceleration benefits.

| Method | Memory | Speedup ↑ | Latency (s) ↓ | LPIPS ↓ | SSIM ↑ | PSNR ↑ |
|---|---|---|---|---|---|---|
| Wan2.1 14B 4bit ($T$=30) | 26.3G | 1× | 241 | - | - | - |
| MagCache-fast | 26.5G | 2.0× | 119 | 0.2223 | 0.6780 | 20.42 |
| MagCache-slow | 26.5G | 1.4× | 169 | 0.1184 | 0.7902 | 22.78 |

## A.4    More Ablations

### A.4.1    Robustness to Denoising Schedulers.

To evaluate the generalizability of the calibrated magnitude ratios across different denoising schedulers, we performed a cross-scheduler validation. Magnitude ratios were calibrated using one scheduler (e.g., UniPC) and then directly applied during inference with a different scheduler (e.g., DPM++). As shown in Table 7, the performance remains remarkably consistent. For instance, ratios calibrated with UniPC and used with DPM++ achieve the same 2.1× speedup and nearly identical SSIM and PSNR scores as when both calibration and inference use DPM++. This demonstrates that the learned magnitude ratios are not specific to the dynamics of a single scheduler and can be generalized effectively.

Table 7: Robustness of magnitude ratios to different schedulers. The speedup and visual quality is stable when the scheduler used for inference differs from the one used for calibration

| Calibrated Scheduler | Inference Scheduler | Speedup ↑ | LPIPS ↓ | SSIM ↑ | PSNR ↑ |
|---|---|---|---|---|---|
| UniPC | UniPC | 2.1× | 0.1053 | 0.8275 | 24.32 |
| UniPC | DPM++ | 2.1× | 0.0976 | 0.8412 | 24.37 |
| DPM++ | DPM++ | 2.1× | 0.0976 | 0.8412 | 24.37 |
| DPM++ | UniPC | 2.1× | 0.1053 | 0.8275 | 24.32 |

### A.4.2    Robustness to the Number of Steps.

We also tested the robustness of the magnitude ratios when the number of inference steps is changed. For these cases, we use nearest-neighbor interpolation to align the calibrated ratio curve to the new number of steps. Table 8 shows that hyperparameters calibrated with 50 steps can be effectively

applied to a 30-step inference process, and vice-versa, without requiring recalibration. The speedup remains constant, and the image quality metrics are determined by the number of inference steps rather than the calibration setting. This stability simplifies the deployment of MagCache, as the pre-calibrated magnitude ratios can be used across different inference steps.

Table 8: Robustness of magnitude ratios to different numbers of inference steps. The speedup is maintained, and quality metrics are consistent for a given number of inference steps, regardless of the number of steps used for calibration.

| Calibrated Steps | Inference Steps | Speedup ↑ | LPIPS ↓ | SSIM ↑ | PSNR ↑ |
|---|---|---|---|---|---|
| 50 | 50 | 2.1× | 0.1053 | 0.8275 | 24.32 |
| 30 | 50 | 2.1× | 0.1053 | 0.8275 | 24.32 |
| 50 | 30 | 2.1× | 0.1917 | 0.7116 | 21.06 |
| 30 | 30 | 2.1× | 0.1917 | 0.7116 | 21.06 |

### A.4.3 Computation of Skip Error in Equation 6

In Section 3.3, we adopt the multiplicative formulation in Equation 6 to compute the single-step skip error $\varepsilon_{\text{skip}}(\hat{t}, t)$ between the cached residual $\mathbf{r}_{\hat{t}}$ at timestep $\hat{t}$ and the ground-truth residual $\mathbf{r}_t$ at timestep $t$. The multiplicative formulation is reasonable according to our following empirical observation and ablation experiments.

**Empirical Observation.** We first define the ground-truth magnitude ratio between the resiudal $\mathbf{r}_t$ and $\mathbf{r}_{\hat{t}}$ as $\Gamma(t, \hat{t})$. Accordding to our empirical observation in Figure 4, the magnitude ratio $\Gamma(t, \hat{t})$ can be approximated by the product $\prod_{i=\hat{t}+1}^{t} \gamma_i$, i.e.:

$$\Gamma(t, \hat{t}) = mean\left(\frac{\|\mathbf{r}_t\|_2}{\|\mathbf{r}_{\hat{t}}\|_2}\right) \approx \prod_{i=\hat{t}+1}^{t} \gamma_i = \prod_{i=\hat{t}+1}^{t} mean\left(\frac{\|\mathbf{r}_i\|_2}{\|\mathbf{r}_{i-1}\|_2}\right). \tag{14}$$

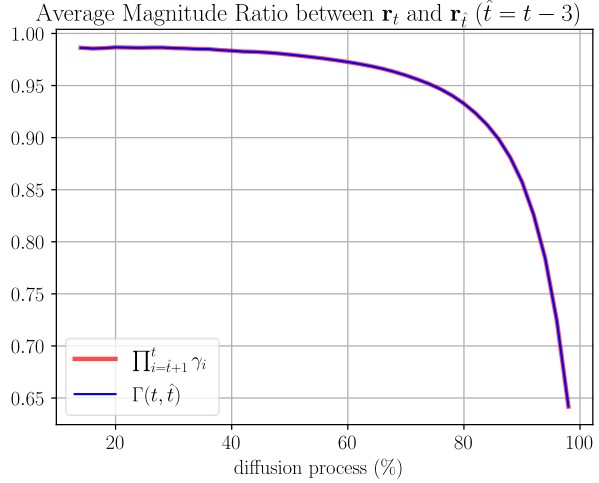

Figure 4: Average Magnitude Ratio between $\mathbf{r}_t$ and $\mathbf{r}_{\hat{t}}$, where $\hat{t} = t - 3$. The $\Gamma(t, \hat{t})$ is the ground-truth magnitude ratio, while the $\prod_{i=\hat{t}+1}^{t} \gamma_i$ is the predicted magnitude ratio using the multiplicative formulation in Equation 6.

Besides, the difference between $\Gamma(t, \hat{t})$ and $\prod_{i=\hat{t}+1}^{t} \gamma_i$ is less than 1e-5 in value. Therefore, the multiplicative formulation in Equation 6 accurately captures the ground-truth magnitude ratio and thus serves as a reliable surrogate.

**Ablation Experiments.** As a naive baseline, we consider a simplified error modeling method that ignores the accumulated error from previously skipped timesteps and considers only the instantaneous magnitude ratio $\gamma_t$ at timestep $t$. The corresponding skip error is defined as:

Table 9: Different error modeling methods of single-step skip error $\varepsilon_{\text{skip}}$ on Wan 2.1. Our multiplicative formulation in Equation 6 performs better than the naive baseline in Equation 15.

| Error Modeling | Latency (s) ↓ | LPIPS ↓ | SSIM ↑ | PSNR ↑ |
|---|---|---|---|---|
| Wan 2.1 | 187 | - | - | - |
| Multiplicative Equation 6 | 87 | **0.1053** | **0.8275** | **24.32** |
| Naive Equation 15 | 84 | 0.1154 | 0.8137 | 24.06 |

$$\varepsilon_{\text{skip}}(\hat{t}, t) = 1 - \gamma_t. \tag{15}$$

As shown in Table 9, our multiplicative formulation (Equation 6) consistently outperforms the naive baseline (Equation 15) across all evaluation metrics. This result aligns with our empirical observation that the multiplicative product $\prod_{i=\hat{t}+1}^{t} \gamma_i$ provides an accurate approximation of the ground-truth magnitude ratio $\Gamma(t, \hat{t})$ between residuals $\mathbf{r}_t$ and $\mathbf{r}_{\hat{t}}$.

*It is also worth noting that when the magnitude ratio exceeds 1.0, we take the absolute value of the skip error, as is done in models like HunyuanVideo and Flux.*

### A.4.4 The Influence of the Initial Steps.

In this section, we investigate the impact of preserving different numbers of initial steps during inference. As shown in Table 10, the first 10 steps are crucial to the overall quality of the generated video. Reducing the number of unchanged initial steps from 10 to 5 leads to a significant degradation in video quality, with LPIPS increasing from 0.1053 to 0.2431, SSIM dropping from 0.8275 to 0.6423, and PSNR falling from 24.32 to 18.80.

While retaining more steps generally improves video quality, it also increases latency and computational cost. To strike a balance between visual fidelity and efficiency, we adopt a default setting that preserves the first 20% of steps, corresponding to 10 steps for Wan 2.1 and 6 steps for Open-Sora.

Table 10: Ablation study on the number of initial unchanged steps for Wan 2.1. The model Wan 2.1 has 50 inference steps in total. †: Default setting where the first 10 steps (20%) are preserved.

| Initial Unchanged Steps | Ratio | Latency (s) ↓ | LPIPS ↓ | SSIM ↑ | PSNR ↑ |
|---|---|---|---|---|---|
| Wan 2.1 $T = 50$ | - | 187 | - | - | - |
| 5 | 10% | 73 | 0.2431 | 0.6423 | 18.80 |
| 10 † | 20% | 87 | 0.1053 | 0.8275 | 24.32 |
| 15 | 30% | 98 | 0.0664 | 0.8966 | 27.71 |

### A.5 Theoretical Analysis of Equation 5

Equation 5 states that when the cosine distance between adjacent residuals is small, the residual difference $||r_t - r_{t-1}||$ can be approximated by the magnitude difference $|||r_t|| - ||r_{t-1}|||$:

$$||r_t - r_{t-1}|| \approx |||r_t|| - ||r_{t-1}|||. \tag{5}$$

This holds when $r_t$ and $r_{t-1}$ are nearly colinear, i.e., $1 - \cos(r_t, r_{t-1}) \approx 0$. Empirically, this occurs in the first 80% of steps. Theoretically, we can analysize the cosine law:

$$||r_t - r_{t-1}||^2 = (||r_t|| - ||r_{t-1}||)^2 + 2||r_t||||r_{t-1}||(1 - \cos(r_t, r_{t-1})). \tag{16}$$

As $1 - \cos(r_t, r_{t-1}) \to 0$, the second term vanishes, yielding Equtaion 5. Let $r(t) = v_\theta(x_t, t) - x_t$. By the chain rule:

$$\frac{dr}{dt} = (\partial_x v_\theta - I)x'(t) + \partial_t v_\theta. \tag{17}$$

Assuming $||x'_t|| \leq M$, spatial lipschitz continuity $||\partial_x v_\theta|| \leq L_x$, and temporal lipschitz continuity $||\partial_t v_\theta|| \leq L_t$, we get

$$\left\|\frac{dr}{dt}\right\| \leq (L_x + 1)M + L_t = B, \tag{18}$$

where $B$ aggregates the ODE's stiffness ($M$) and model smoothness ($L_x, L_t$), which is usually within a small constant range. Integrating over step size $h$(small in early steps), we can get $||r_t - r_{t-1}|| \leq Bh$. Assuming $||r_t||, ||r_{t-1}|| \geq c > 0$, we get:

$$1 - \cos(r_t, r_{t-1}) = \frac{||r_t - r_{t-1}||^2 - (||r_t|| - ||r_{t-1}||)^2}{2||r_t|| ||r_{t-1}||} \leq \frac{||r_t - r_{t-1}||^2}{2||r_t|| ||r_{t-1}||} \leq \frac{B^2 h^2}{2c^2} = O(h^2). \tag{19}$$

Hence, $r_t$ and $r_{t-1}$ are nearly aligned when $h$ is small. In non-uniform schedules (e.g., shift=8), early steps use small $h$ (e.g., 0.001), making $1 - \cos(r_t, r_{t-1})$ negligible, validating Equation 5.

### A.6 More Visualization Cases

In this section, we present additional qualitative results, including both videos and images, to further demonstrate the effectiveness of MagCache. Compared with TeaCache, MagCache consistently achieves superior visual quality while maintaining comparable or lower latency. Specifically, Mag-Cache consistently delivers better alignment with ground-truth content, improved preservation of fine visual details, and enhanced rendering of textual elements, such as clearer and more accurate text generation in both videos and images. The qualitative results span four widely used video generation models and one state-of-the-art image generation model: Wan 2.1 1.3B in Figure 5, Wan 2.1 14B in Figure 6, Open-Sora in Figure 7, HunyuanVideo in Figure 8, and Flux(Image Generation Model) in Figure 9.

**Wan 2.1 1.3B**

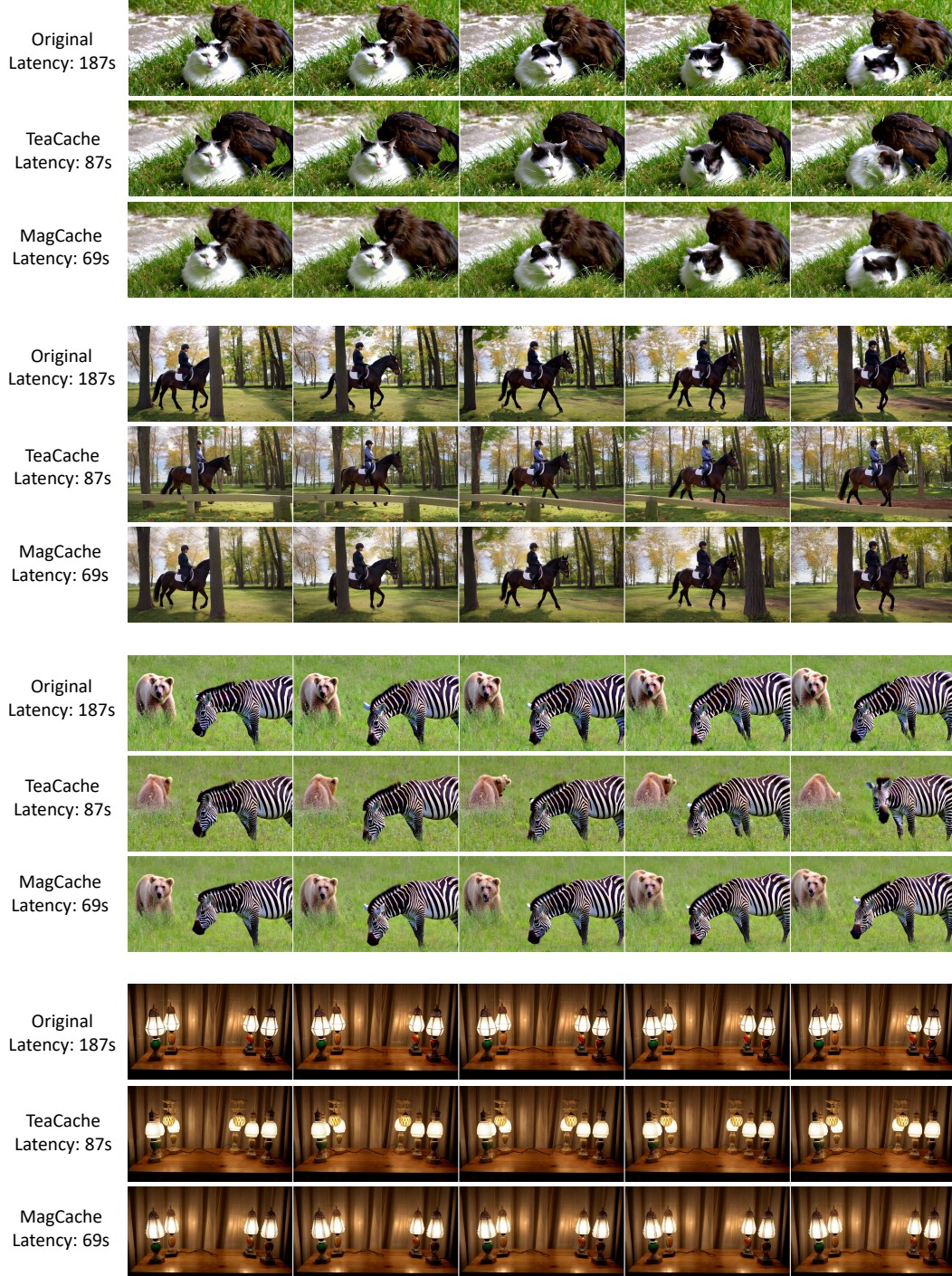

Figure 5: Videos generated by Wan 2.1 1.3B using original model, Teacache-Fast, and our MagCache-Fast. Best-viewed with zoom-in.

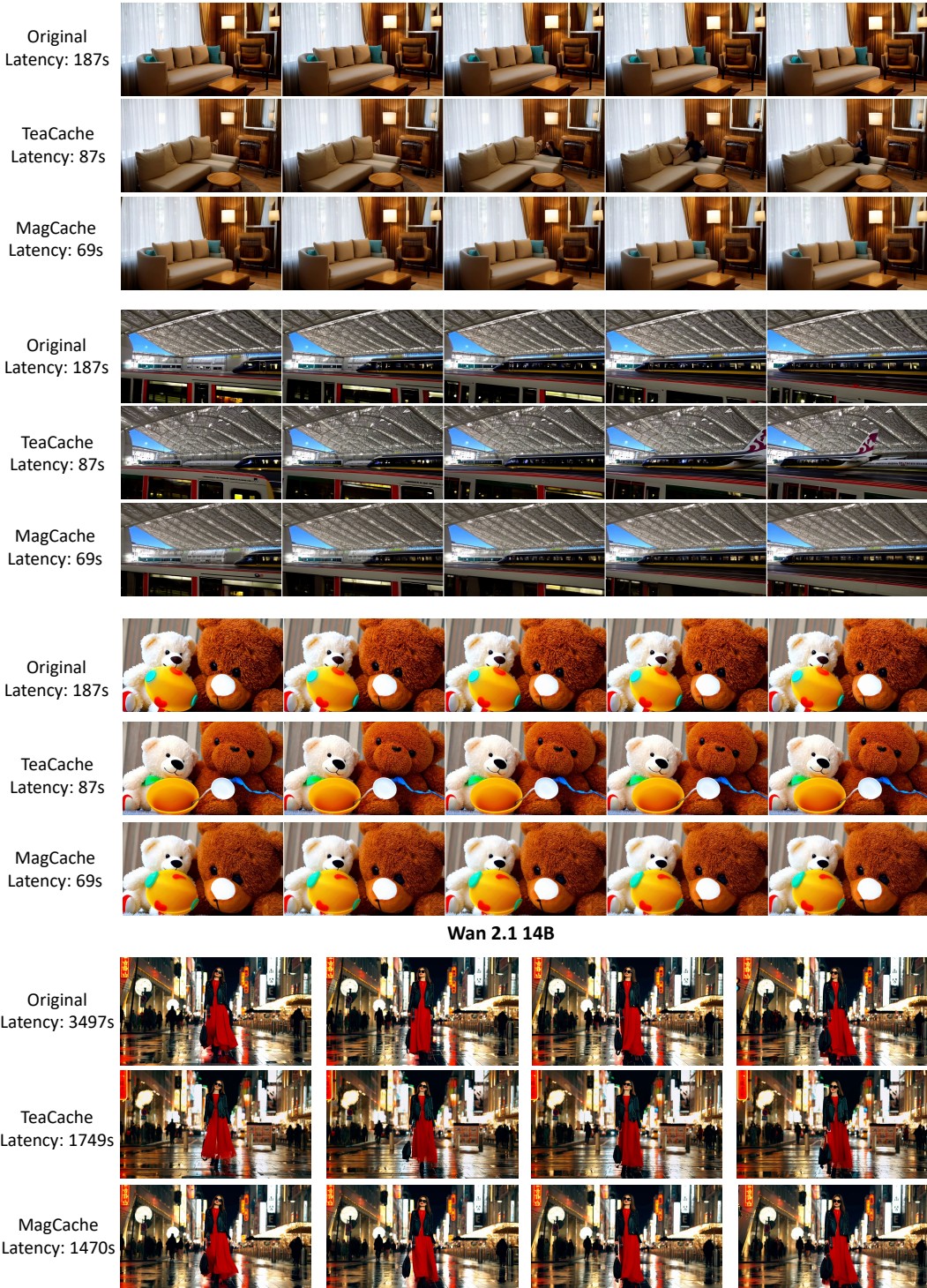

Figure 6: Videos generated by Wan 2.1 1.3B and Wan 2.1 14B using original model, Teacache-Fast, and our MagCache-Fast. Best-viewed with zoom-in.

**Open-Sora**

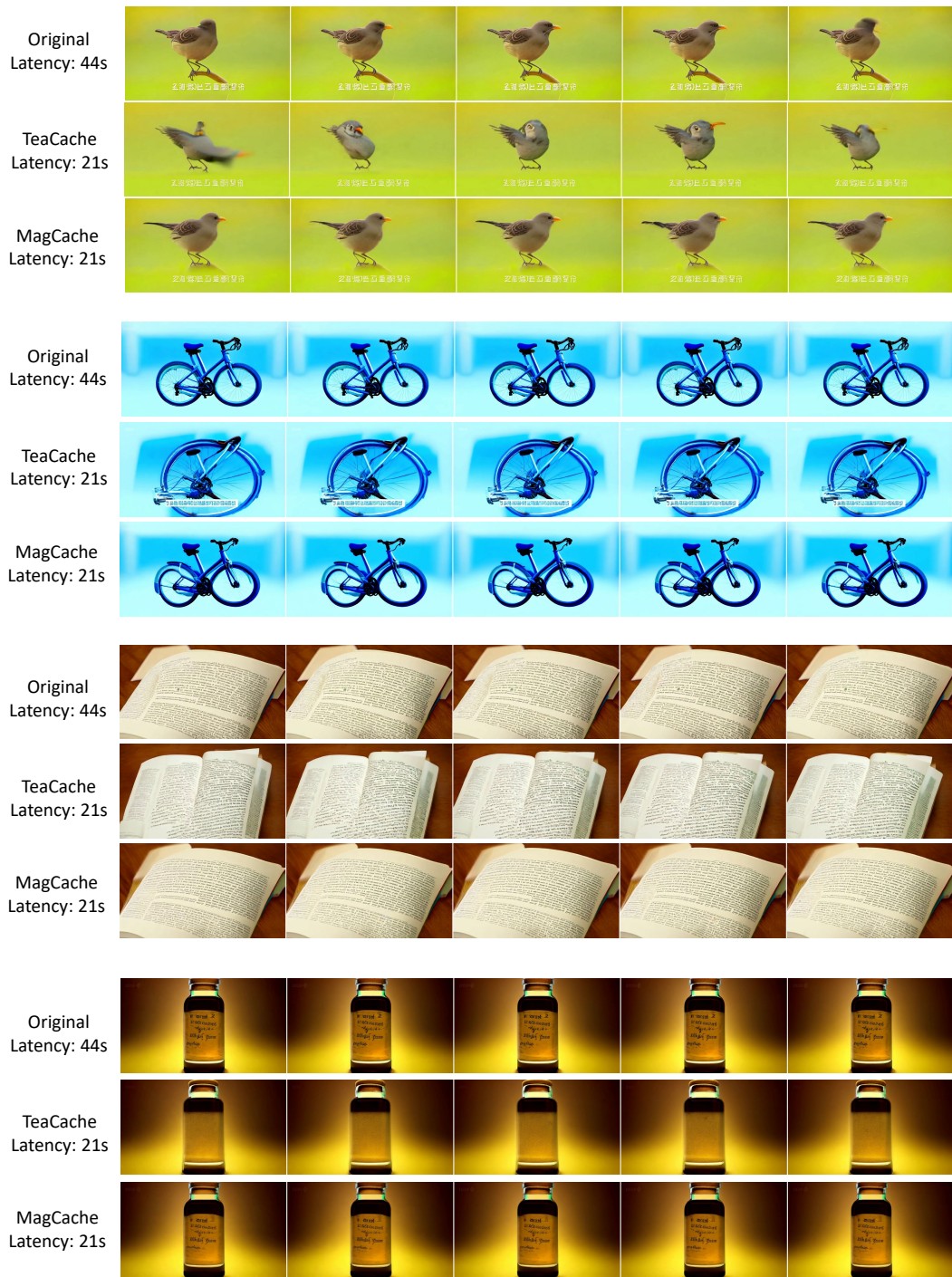

Figure 7: Videos generated by Open-Sora using original model, Teacache-Fast, and our MagCache-Fast. Best-viewed with zoom-in.

**HunyuanVideo**

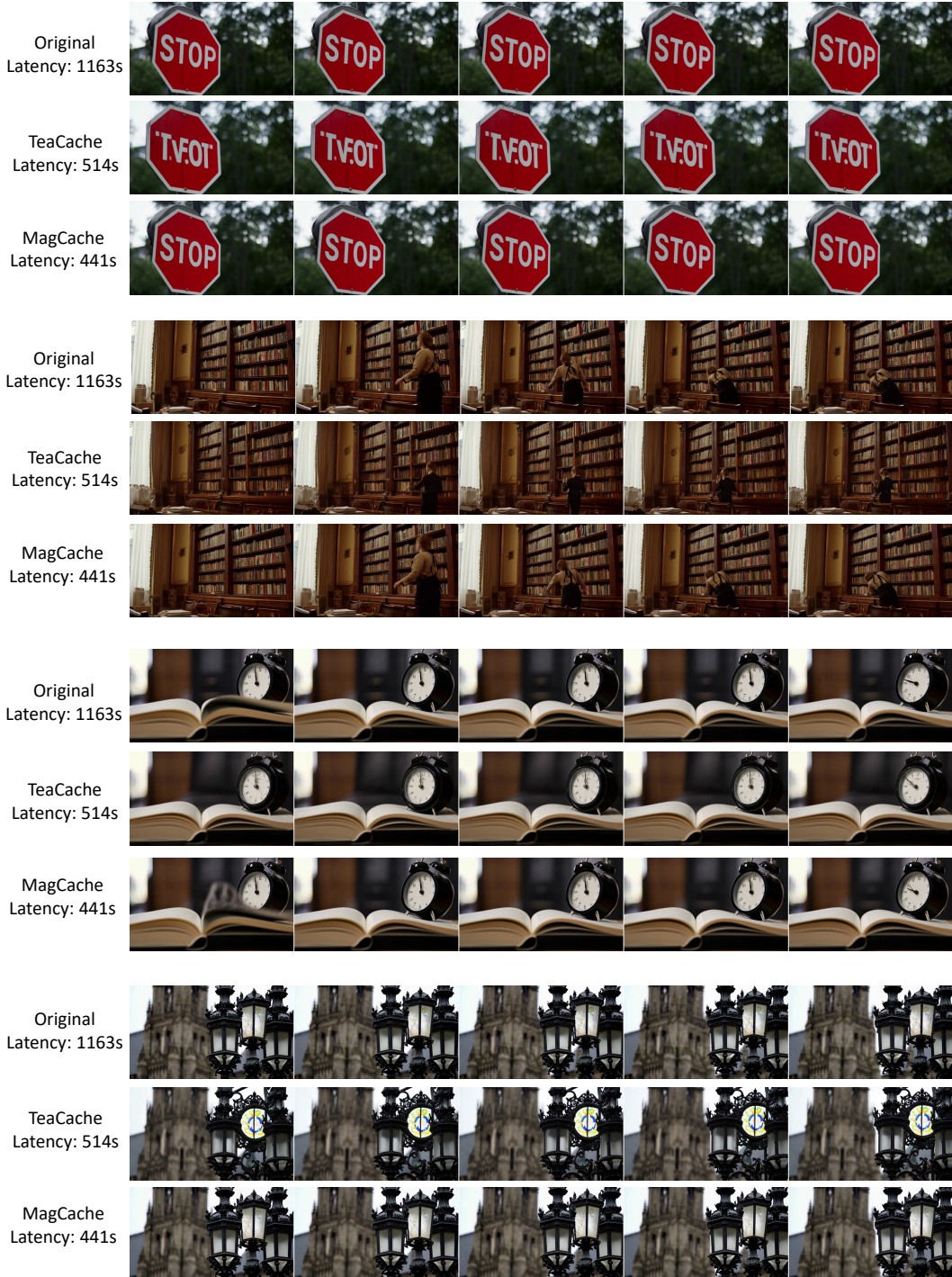

Figure 8: Videos generated by HunyuanVideo using original model, Teacache-Fast, and our MagCache-Fast. Best-viewed with zoom-in.

**Flux**

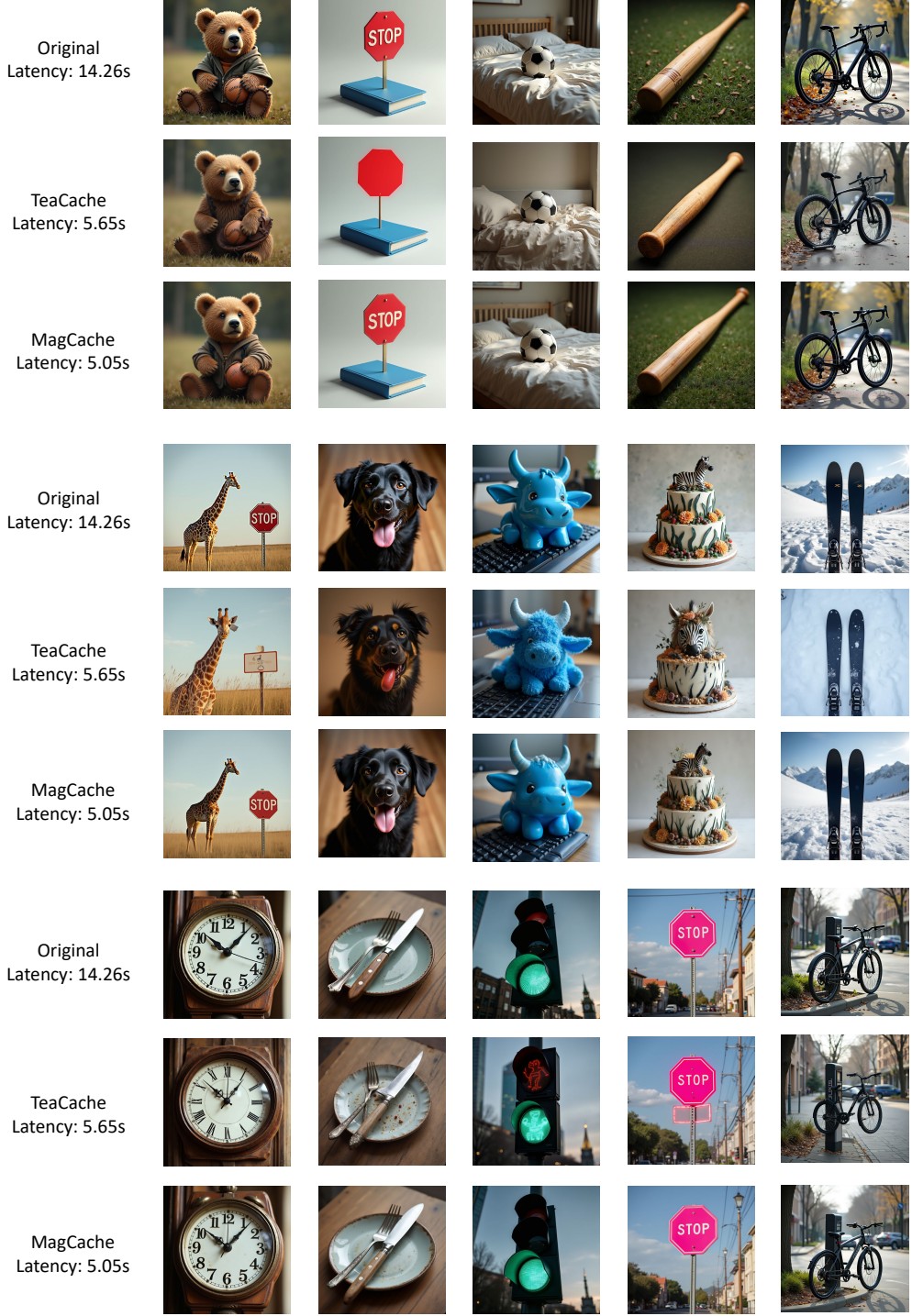

Figure 9: Images generated by Flux using original model, Teacache-Fast, and our MagCache-Fast. Best-viewed with zoom-in.

