# OpenReview forum: "MagCache: Fast Video Generation with Magnitude-Aware Cache"
_NeurIPS.cc/2025/Conference — NeurIPS 2025 poster_

### Official Review · Reviewer_YTog · 2025-06-29

**Clarity:** 3
**Significance:** 2
**Originality:** 3
**Rating:** 4
**Confidence:** 4

**Summary:**

This paper introduces MagCache, a novel caching strategy to accelerate video generation in diffusion models. MagCache leverages a newly discovered law: the magnitude ratio of successive residual outputs decreases monotonically across different models and prompts. By employing an error modeling mechanism and adaptive caching strategy, it adaptively skips unimportant timesteps, thereby significantly boosting inference speed while maintaining high visual fidelity.

**Questions:**

See weakness

**Ethical Concerns:**

["NO or VERY MINOR ethics concerns only"]

**Limitations:**

yes

**Quality:**

2

**Strengths And Weaknesses:**

Strengths：

1. This paper proposed a novel unified magnitude decay law that reveals the stability and monotonicity of the residual magnitude ratios in video diffusion models, providing a theoretical basis for skipping timesteps.

2. MagCache performs excellently on multiple benchmark models (such as Open-Sora and Wan 2.1), achieving significant acceleration (up to 2.68×) and superior visual quality compared to existing methods.

3. MagCache can be seamlessly integrated into existing diffusion model pipelines without requiring additional training or large amounts of data.

4. The writing of this paper is clear and easy to follow.


Weaknesses&Question：

Compatibility with other acceleration methods: Can MagCache be integrated with other diffusion model acceleration techniques, such as model distillation and low-precision arithmetic?

---

> ### Author Rebuttal · Authors · 2025-07-30
>
> Thanks for the valuable feedback. **MagCache is a unified, plug-and-play acceleration method compatible with a wide range of video generation models**, including Wan 2.1, Open-Sora, HunyuanVideo, CogVideoX, few-step distilled models (FusionX), and low-precision models. It supports ComfyUI interface for ease of use.
>
> ---
>
> **Q1:** Compatibility with other acceleration methods: Can MagCache be integrated with other diffusion model acceleration techniques, such as model distillation and low-precision arithmetic?
>
> **A1:** Thanks for the insightful question. **MagCache is compatible with other acceleration techniques, including model distillation and low-precision arithmetic.**
>
> Although distilled models tend to exhibit larger variations between denoising steps, **MagCache remains effective even for few-step distilled models by caching and reusing intermediate computations with minimal differences.** As shown in Table R4.1, we evaluated MagCache on FusionX (a 10-step distilled version of Wan2.1 14B) and achieved a 1.66× speedup without significant degradation in visual quality. Since distillation has reduced redundancy between steps, the speedup ratio of MagCache (1.66×) on the distilled model (FusionX) is lower than that on the non-distilled version (Wan 2.1). Notably, simply reducing the inference steps to T=6 yields much worse performance than using MagCache with T=10, even at the same speedup of 1.66×.
>
> We also evaluated MagCache under low-precision settings. **As shown in Table R4.2, MagCache is compatible with 4-bit quantized models**, maintaining acceleration benefits with only a minor drop in speedup due to the reduced numerical precision. Both the “fast” and “slow” variants maintain strong visual quality when used with low-precision models.
>
> ---
>
> **Table R4.1: Evaluation of MagCache on Few-Step Distilled Model (Wan 2.1 14B FusionX, 33 frames, 480P)**
>
> | **Method**                                         | **Speedup ↑** | **Latency (s) ↓** | **LPIPS ↓** | **SSIM ↑** | **PSNR ↑** |
> | -------------------------------------------------- | ------------- | ----------------- | ----------- | ---------- | ---------- |
> | Wan2.1 14B FusionX (T = 10)                        | 1×            | 30                | -           | -          | -          |
> | Wan2.1 14B FusionX (T = 6)                         | 1.66x         | 18                | 0.2982      | 0.6471     | 20.35      |
> | Wan2.1 14B FusionX + MagCache (T=10, skip 4 steps) | **1.66×**     | **18**            | **0.1812**  | **0.7868** | **24.23**  |
>
> ---
>
> **Table R4.2: Evaluation of MagCache on Low-precision Arithmetic Model (Wan 2.1 14B, 33 frames, 480P)**
>
> | **Method**                             | Memory | **Speedup ↑** | **Latency (s) ↓** | **LPIPS ↓** | **SSIM ↑** | **PSNR ↑** |
> | -------------------------------------- | ------ | ------------- | ----------------- | ----------- | ---------- | ---------- |
> | Wan2.1 14B 4bit (T = 30)               | 26.3G  | 1×            | 241               | -           | -          | -          |
> | Wan2.1 14B 4bit (T = 30)+MagCache-fast | 26.5G  | 2.0x          | 119               | 0.2223      | 0.6780     | 20.42      |
> | Wan2.1 14B 4bit (T = 30)+MagCache-slow | 26.5G  | 1.4x          | 169               | 0.1184      | 0.7902     | 22.78      |

---

> > ### Comment · Reviewer_YTog · 2025-08-05
> >
> > Thank you for your response. I would like to maintain my original score.

---

> ### Author Response · Authors · 2025-08-08
>
> Dear Reviewer YTog,
>
> Thank you sincerely for your valuable time to review our paper and providing such insightful feedback. We deeply appreciate the thought and effort you have invested in this process. We wanted to update you that the **reviewer AVyF and 9jpp , who initially assigned scores of 3, have not yet joined any discussion.** Given this, we would be extremely grateful if you might be willing to reconsider our work with a higher score.
>
> **Any slight adjustment to your score, or your continued support when engaging in discussions with the Area Chair, would mean a great deal to us as we strive to address feedback and strengthen our submission.**
>
> Thank you again for your support and consideration.
>
> Sincerely,
>
> Authors of Paper 16663

---

### Official Review · Reviewer_9jpp · 2025-06-29

**Clarity:** 3
**Significance:** 3
**Originality:** 2
**Rating:** 4
**Confidence:** 4

**Summary:**

This paper presents MagCache, a novel magnitude-aware caching strategy that efficiently accelerates video generation in diffusion models with minimal calibration cost. Extensive experiments on models like Open-Sora and Wan 2.1 demonstrate its effectiveness, and the method shows potential as a plug-and-play solution for faster inference.

**Questions:**

1. How should δ and K be selected when applying MagCache to other video generation models? Are there any heuristics or data-driven methods that guide this choice? Is the cost high?
2. How does the type of video content (e.g., human, natural scene, dynamic textures) influence the selection of γ and the effectiveness of MagCache?
3. How sensitive is MagCache to different prompt types? Does prompt complexity or semantic category affect caching behavior or performance?

**Ethical Concerns:**

["NO or VERY MINOR ethics concerns only"]

**Final Justification:**

The authors’ rebuttal has resolved some of my earlier concerns and provided helpful clarifications. Accordingly, I am revising my score from borderline reject to borderline accept.

**Limitations:**

Yes. The authors have clearly acknowledged the limitations of their approach and briefly discussed potential negative societal impacts. Their transparency is appreciated.

**Paper Formatting Concerns:**

No major formatting issues observed. The paper appears to follow the NeurIPS 2025 formatting guidelines.

**Quality:**

3

**Strengths And Weaknesses:**

Strengths:
1. The paper proposes MagCache, a novel magnitude-aware caching mechanism designed to accelerate video generation in diffusion models.
2. Extensive experiments demonstrate that MagCache achieves notable speedups on state-of-the-art video diffusion models such as Open-Sora (2.1×) and Wan 2.1 (2.68×), showcasing its practical effectiveness.
3. The method requires only a single random sample for calibration, significantly reducing both computational overhead and calibration time.

Weaknesses:
1. Insufficient motivation and generality. While MagCache aims to accelerate inference by exploiting motion-aware magnitude patterns, the empirical motivation presented in Section 3.2 is not fully convincing. In video generation, various motion types, magnitudes, and semantic content (e.g., human vs. landscape) can significantly affect inference dynamics. However, the current analysis relies only on two video models (Open-Sora, Wan 2.1) and three prompts, which is limited in scope.

1.1 The choice of models is too narrow to support the claim that MagCache is a \textbf{unified}, plug-and-play solution.

1.2 Figure 1 uses only three prompts, which is insufficient to substantiate the prompt-invariance claim made in Section 3.2.

2. Lack of theoretical insight into Equation (5). While the empirical benefits of the magnitude ratio are shown, the underlying mechanism and theoretical justification remain under-explored. A deeper analysis would strengthen the motivation and reveal how MagCache influences the denoising dynamics at a more principled level.

3. Parameter sensitivity and lack of consistency. In Section 3.3.2, critical parameters δ and K are introduced without adequate justification or guidance. Moreover, the settings for δ and K vary across experiments (e.g., K = 1, 2, 3, 4), indicating a lack of standardized configuration. The absence of a consistent selection rule raises concerns about reproducibility, robustness, and generalization.

4. Figure 2 is difficult to interpret and lacks clarity in both visual design and explanation.

---

> ### Author Rebuttal · Authors · 2025-07-30
>
> **Q1:** Insufficient motivation and generality. While MagCache aims to accelerate inference by exploiting motion-aware magnitude patterns, the empirical motivation is not fully convincing. The current analysis on two video models and three prompts is limited in scope.
>
> **A1:** Thanks for the insightful feedback. **Our core motivation is to accelerate video generation by exploiting redundancy in residuals across adjacent steps.** Emperically, the residual difference is small in the first 80% steps across diverse models and prompts. Notably, MagCache is step-aware, not motion-aware. Both motion and semantics evolve minimally between early adjacent steps. **Besides, we extend experiments to include more models (Table R3.1) and a broader set of prompts (Table R3.2 & R3.3) to demonstrate the effectiveness and generality of MagCache.**
>
>
>
> **Q1.1:** Model diversity is limited to support the plug-and-play claim.
>
> **A1.1:** To strengthen the generality, we integrate MagCache to HunyuanVideo, CogVideoX, and Flux (Table R3.1), observing competitive or superior performance across these models. We also qualitatively validated MagCache on additional models (FramePack, VACE, Flux Kontext, Chroma, and OmniGen2), and will release supporting code and demos. **In summary, MagCache is a unified, plug-and-play acceleration method applicable to a wide range of diffusion models.**
>
>
>
> **Table R3.1 Generalization of MagCache across Diverse Diffusion Models.**
>
> | Method                                      | Speedup ↑ | SSIM ↑ | PSNR ↑ |
> | ------------------------------------------- | --------- | ------ | ------ |
> | **HunyuanVideo (T = 50, 129 frames, 540P)** | 1         | -      | -      |
> | TeaCache-slow                               | 1.6       | 0.787  | 23.8   |
> | TeaCache-fast                               | 2.2       | 0.774  | 23.3   |
> | MagCache-slow                               | 2.2       | 0.945  | 34.5   |
> | MagCache-fast                               | 2.6       | 0.920  | 31.7   |
> | **CogVideoX2B (T = 50, 49 frames, 480P)**   | 1         | -      | -      |
> | TeaCache                                    | 2.3       | 0.881  | 27.0   |
> | MagCache                                    | 2.3       | 0.921  | 30.4   |
> | **Flux (T = 28)**                           | 1         | -      | -      |
> | TeaCache-slow                               | 2.0       | 0.774  | 20.1   |
> | TeaCache-fast                               | 2.5       | 0.702  | 18.1   |
> | MagCache-slow                               | 2.5       | 0.888  | 24.4   |
> | MagCache-fast                               | 2.8       | 0.809  | 21.3   |
>
> ---
>
> **Q1.2:** Figure 1 uses only three prompts,  insufficient to substantiate the prompt-invariance.
>
> **A1.2:** To assess prompt invariance, we analyzed magnitude ratios over **944 VBench prompts**, grouped into five 20% diffusion intervals. **As shown in Table R3.2, the magnitude ratios exhibit low standard deviation and a consistent decreasing trend,** with a sharp drop in the final 20%. Cosine distances follow a similar pattern, aligning with Figure 1.
>
> Further, Table R3.3 demonstrates that the choice of calibration prompt has negligible effect. Whether using a random prompt, the full prompt set, or an outlier, performance remains stable. **In summary, magnitude ratios are consistent across prompts, and MagCache is robust to prompt choice.**
>
>
>
> **Table R3.2 Statistics of Magnitude Ratios and Cosine Distance across 944 Vbench prompts.** `0.9965 (± 0.0004) denotes mean ± standard deviation`
>
> | Diffusion Process (%) | Magnitude Ratio (Wan2.1 ) | Cosine Distance (Wan2.1) |
> | --------------------- | ------------------------- | ------------------------ |
> | 0–20                  | 0.9965 (±0.0004)          | 0.0008 (±0.0003)         |
> | 20–40                 | 0.9951 (±0.0002)          | 0.0003 (±0.0003)         |
> | 40–60                 | 0.9928 (±0.0002)          | 0.0001 (±0.0001)         |
> | 60–80                 | 0.9851 (±0.0002)          | 0.0002 (±0.0001)         |
> | 80–100                | 0.9285 (±0.0019)          | 0.0054 (±0.0009)         |
>
> **Table R3.3 Effect of Calibration Prompt.**
>
> | Calibration Prompt                     | **Speedup ↑** | **SSIM ↑** | **PSNR ↑** |
> | -------------------------------------- | ------------- | ---------- | ---------- |
> | Random Prompt (Ours)                   | 2.1           | 0.813      | 23.4       |
> | 944 Prompts (Average Magnitudes)       | 2.1           | 0.816      | 23.5       |
> | Outlier Prompt (Farthest from Average) | 2.1           | 0.810      | 23.3       |
>
> ---
>
> **Q2:** Lack of theoretical insight into Equation (5).
>
> **A2:** Thanks for the valuable comment. Equation (5) states that when the cosine distance between adjacent residuals is small, the residual difference $||r\_t - r\_{t-1}||$ can be approximated by the magnitude difference $\left|||r\_t|| - ||r\_{t-1}||\right|$:
> $$
> ||r_t - r_{t-1}|| \approx \left|||r_t|| - ||r_{t-1}||\right|. \tag{5}
> $$
>
> This holds when $r\_t$ and $r\_{t-1}$ are nearly colinear (i.e., $1 - \cos(r\_t, r\_{t-1}) \approx 0$). Empirically, this occurs in the first 80% of steps. Theoretically, we can analysize the cosine law:
>
> $$
> ||r_t - r_{t-1}||^2 = (||r_t|| - ||r_{t-1}||)^2 + 2||r_t||||r_{t-1}||(1 - \cos(r_t, r_{t-1})).
> $$
>
> As $1 - \cos(r\_t, r\_{t-1}) \to 0$, the second term vanishes, yielding Eq. (5). Let $r(t) = v\_\theta(x\_t, t) - x\_t$. By the chain rule:
> $$
> \frac{dr}{dt} = (\partial_x v_\theta - I) x'(t) + \partial_t v_\theta.
> $$
> Assuming $||x'\_t|| \le M$, spatial lipschitz continuity $||\partial\_x v\_\theta|| \le L\_x$, and temporal lipschitz continuity $||\partial\_t v\_\theta|| \le L\_t$, we get
>
>
>
> $$
> \left|\left|\frac{dr}{dt}\right|\right| \le (L_x + 1)M + L_t = B,
> $$
>
> where $B$ aggregates the ODE’s stiffness ($M$) and model smoothness ($L_x, L_t$), which is usually within a small constant range. Integrating over step size $h$(small in early steps), we can get $||r_t - r_{t-1}|| \le Bh.$ Assuming $||r\_t||, ||r\_{t-1}|| \ge c > 0$, we get:
>
> $$
> 1 - \cos(r_t, r_{t-1}) = \frac{||r_t - r_{t-1}||^2 -(||r_t|| - ||r_{t-1}||)^2}{2||r_t||||r_{t-1}||}\le \frac{||r_t - r_{t-1}||^2}{2||r_t||||r_{t-1}||} \le \frac{B^2 h^2}{2c^2} = O(h^2).
> $$
>
> Hence, $r\_t$ and $r\_{t-1}$ are nearly aligned when $h$ is small. In non-uniform schedules (e.g., shift=8), early steps use small $h$ (e.g., 0.001), making $1 - \cos(r\_t, r\_{t-1})$ negligible, validating Equation (5).
>
> ---
>
> **Q3:** Parameter sensitivity and lack of consistency.
>
> **A3:** The varying $\delta$ and $K$ settings across models were chosen to ensure fair comparisons at similar speedups. In practice, **MagCache performs robustly with default parameters across different models**, requiring only minor $\delta$ adjustments. We offer two predefined modes:
>
> - **Slow mode: $K = 2$, $\delta = 0.06$ (default)**
> - **Fast mode: $K = 4$, $\delta = 0.06$ (default)**
>
> Here, $K$ controls speedup mode, and $\delta$ adjusts the quality–speed trade-off within the selected mode. In Table R1.1, varying $\delta$ offers fine control with just 1–2 adjustments (doubling or halving). **We recommend the slow mode ($K=2, \delta=0.06$), which works well for various models.** **Our ComfyUI interface allows users to easily adjust these parameters in a few seconds.**
>
>
>
> **Table R3.4 Parameter Configurations in Slow and Fast inference mode.**
>
> | **Mode**                     | $K$   | $\delta$ | **Speedup ↑** | **SSIM ↑** | **PSNR ↑** |
> | :--------------------------- | ----- | -------- | ------------- | ---------- | ---------- |
> | **MagCache-slow (Wan2.1)**   | **2** | **0.06** | 2.0           | 0.838      | 24.5       |
> |                              |       | 0.12     | 2.1           | 0.8275     | 24.3       |
> |                              |       | 0.03     | 1.9           | 0.842      | 24.6       |
> | **MagCache-fast (Wan2.1)**   | **4** | **0.06** | 2.4           | 0.774      | 22.3       |
> |                              |       | 0.12     | 2.7           | 0.757      | 22.2       |
> |                              |       | 0.03     | 2.0           | 0.782      | 22.5       |
> | **MagCache-slow (OpenSora)** | **2** | **0.06** | 1.8           | 0.814      | 23.9       |
> |                              |       | 0.12     | 1.9           | 0.813      | 23.8       |
> |                              |       | 0.03     | 1.7           | 0.816      | 24.0       |
> | **MagCache-fast (OpenSora)** | **4** | **0.06** | 2.1           | 0.763      | 21.9       |
> |                              |       | 0.12     | 2.4           | 0.754      | 21.7       |
> |                              |       | 0.03     | 2.0           | 0.766      | 22.0       |
>
> ---
>
> **Q4:** Figure 2 is difficult to interpret and lacks clarity in both visual design and explanation.
>
> **A4:** We have revised Figure 2 for improved clarity. It now better illustrates how $K$ and $\delta$ jointly govern the adaptive caching and reusing, with annotations and a step-wise flow to highlight the caching logic. Due to the rebuttal policy, we will update the figure in the revised paper.
>
> ---
>
> **Q5:** How does the type of video content (e.g., human, natural scene, dynamic textures) influence the the effectiveness of MagCache? How sensitive is MagCache to different prompt types?
>
> **A5:** We group the 944 Vbench prompts into 5 types. Table R3.5 shows that prompt types have negligible effect on MagCache’s performance. Stable magnitude ratios across prompts (Table R3.2) and calibration robustness (Table R3.3) also support this conclusion.
>
> **Table R3.5 Robustness of MagCache across Different Prompt Types.**
>
> | Prompt Type     | **SSIM ↑** | **PSNR ↑** |
> | --------------- | ---------- | ---------- |
> | Human           | 0.824      | 23.4       |
> | Objects         | 0.812      | 23.5       |
> | Natural Scene   | 0.812      | 23.2       |
> | Dynamic Motion  | 0.805      | 23.0       |
> | Fictional Scene | 0.828      | 23.4       |
> | Average         | 0.813      | 23.4       |

---

> ### Author Response · Authors · 2025-08-06
>
> Dear Reviewer 9jpp,
>
> Thank you for your valuable time in reviewing our manuscript. As the discussion phase is ending soon,  we would greatly appreciate it if you can consider our response and supplementary experiments in your final assessment. We demonstrated the robustness and generality of MagCache across more models and prompts (Table R3.1-R3.3, R3.5), and included a theoretical analysis of Equation (5) in A2. Additionally, we have improved the clarity of Figure 2, and shown that the parameters of MagCache are consistent, robust, and easy to set (Table R3.4). We hope that our response and the supplementary experiments have satisfactorily addressed your concerns.
>
> Thank you for your valuable time!
>
> Sincerely,
>
> Authors of Paper 16663

---

> ### Author Response · Authors · 2025-08-08
>
> Dear Reviewer 9jpp,​
>
> I hope this message finds you well. Please accept my sincere apologies for following up again; we understand your time is extremely valuable. However, I’m writing with urgency as tomorrow marks the final day of our paper’s rebuttal review period, and we haven’t yet received your feedback on our rebuttal.​
>
> We’ve worked diligently to address all concerns from your initial review, hoping our rebuttal clarifies our work. With only one day left until the deadline, we’d be deeply grateful if you could spare a few moments to review it at your convenience. Your insights are critical to our paper’s evaluation at this stage, and we hope our rebuttal has adequately addressed the points you raised before the deadline closes.​
>
> Thank you again for your time and consideration. We apologize for any inconvenience and look forward to your input.​
>
> Sincerely,
>
> Authors of Paper 16663

---

### Official Review · Reviewer_AVyF · 2025-07-02

**Clarity:** 3
**Significance:** 3
**Originality:** 3
**Rating:** 4
**Confidence:** 3

**Summary:**

This paper proposes MagCache, a training-free method to accelerate video diffusion models by adaptively skipping redundant timesteps based on a newly observed magnitude decay law. The authors find that the L2 norm of residuals between timesteps decreases consistently, enabling a reliable and prompt-agnostic criterion for caching. MagCache models skip error using this magnitude ratio and reuses cached features when the error is within a threshold. Experiments on Open-Sora and Wan 2.1 show 2.1×–2.7× speedup with better or comparable visual quality than prior methods like TeaCache.

**Questions:**

See weaknesses.

**Ethical Concerns:**

["NO or VERY MINOR ethics concerns only"]

**Final Justification:**

The author has effectively addressed my concerns, and I have decided to increase my score.

**Limitations:**

yes

**Quality:**

3

**Strengths And Weaknesses:**

Strengths

- This paper identifies a magnitude decay law that is consistent across models and prompts.
- MagCache requires only a single calibration sample, no retraining, and integrates easily.
- MagCache achieves 2×+ speedups while improving visual metrics (LPIPS, SSIM, PSNR).

Weaknesses
- A notable weakness of the paper is the incompleteness of baseline comparisons. In particular, the authors do not compare MagCache against two recent state-of-the-art diffusion acceleration methods: TaylorSeer[1] and DuCa[2].
- The ~2× speedup is solid but not as high as some recent approaches (e.g., TaylorSeer ~5×).
- The experiments are conducted on two large text-to-video diffusion models (Open-Sora and Wan), but broader validation is somewhat limited. The method’s generality would be better established by testing on more diverse setups (e.g., CogVideoX).


Reference:
- [1] Accelerating Diffusion Transformers with Dual Feature Caching, ICCV 2025.
- [2] DuCa: Accelerating Diffusion Transformers with Dual Feature Caching, arxiv 2024.

---

> ### Author Rebuttal · Authors · 2025-07-30
>
> **Q1:** Comparison with two recent state-of-the-art diffusion acceleration methods: TaylorSeer[1] and DuCa[2]
>
> **A1:** Thanks for the valuable suggestion. We have added detailed comparisons with TaylorSeer [1] and DuCa [2] across three representative models: Open-Sora, Wan2.1 1.3B, and HunyuanVideo in Table R2.1. A comparison will be included in the revised paper.
>
> Compared with DuCa on Open-Sora and HunyuanVideo, MagCache achieves better visual quality at similar speedups, while requiring only 0.1 GB of extra memory—significantly less than DuCa’s 23.9 GB (Open-Sora) and 14.7 GB (HunyuanVideo). Against TaylorSeer, MagCache delivers better visual quality on both Wan2.1 1.3B and HunyuanVideo at similar speedups, with much lower memory overhead (0.1GB-0.5 GB vs. 40–46 GB).
>
> It is worth noting that TaylorSeer and DuCa cache outputs of all transformer layers, while MagCache only caches a single residual output, greatly reducing memory usage.
> **In summary, MagCache consistently outperforms TaylorSeer and DuCa in both visual quality and memory efficiency.**
>
>
>
> **Table R2.1: Comparison with Other Cache-based Methods across Various Models.**
>
> | **Method**                          | **Speedup ↑** | Extra Memory↓ | **Latency (s) ↓** | **LPIPS ↓** | **SSIM ↑** | **PSNR ↑** |
> | ----------------------------------- | ------------- | ------------- | ----------------- | ----------- | ---------- | ---------- |
> | **Open-Sora 1.2** (51 frames, 480P) |               |               |                   |             |            |            |
> | Open-Sora 1.2 (T = 30)              | 1×            | -             | 44.56             | -           | -          | -          |
> | DuCa                                | 2.08×         | 23.9G         | 21.42             | 0.2316      | 0.7652     | 19.96      |
> | TeaCache-slow                       | 1.40×         | **0.1G**      | 31.69             | 0.1303      | 0.8405     | 23.67      |
> | TeaCache-fast                       | 2.05×         | **0.1G**      | 21.67             | 0.2527      | 0.7435     | 18.98      |
> | MagCache-slow                       | 1.41×         | **0.1G**      | 31.48             | **0.0827**  | **0.8859** | **26.93**  |
> | MagCache-fast                       | 2.10×         | **0.1G**      | 21.21             | 0.1522      | 0.8266     | 23.37      |
> | **Wan 2.1 1.3B** (81 frames, 480P)  |               |               |                   |             |            |            |
> | Wan 2.1 (T = 50)                    | 1×            |               | 187.21            | -           | -          | -          |
> | TeaCache-slow                       | 1.59×         | **0.5G**      | 117.20            | 0.1258      | 0.8033     | 23.35      |
> | TeaCache-fast                       | 2.14×         | **0.5G**      | 87.55             | 0.2412      | 0.6571     | 18.14      |
> | TaylorSeer(N=2, O=1)                | 2.07×         | 40.0G         | 90.15             | 0.3792      | 0.5220     | 15.06      |
> | MagCache-slow                       | 2.14×         | **0.5G**      | 87.27             | **0.1206**  | **0.8133** | **23.42**  |
> | MagCache-fast                       | 2.68×         | **0.5G**      | 69.75             | 0.1748      | 0.7490     | 21.54      |
> | **Hunyuan Video** (65 frames, 480P) |               |               |                   |             |            |            |
> | Hunyuan Video (T=50)                | 1×            |               | 177.50            | -           | -          | -          |
> | DuCa                                | 4.62×         | 14.7G         | 38.41             | 0.3516      | 0.6443     | 16.16      |
> | TeaCache-slow                       | 2.97×         | **0.1G**      | 59.76             | 0.1759      | 0.7927     | 20.23      |
> | TeaCache-fast                       | 4.55×         | **0.1G**      | 39.01             | 0.4377      | 0.6216     | 16.07      |
> | TaylorSeer (N=3, O=1)               | 3.00×         | 46.2G         | 59.16             | 0.1353      | 0.8335     | 21.42      |
> | TaylorSeer (N=5, O=1)               | 5.00×         | 46.2G         | 35.50             | 0.2691      | 0.7039     | 16.79      |
> | MagCache-slow                       | 3.10×         | **0.1G**      | 57.20             | **0.1283**  | **0.8372** | **24.38**  |
> | MagCache-fast                       | 5.00×         | **0.1G**      | 35.45             | 0.2656      | 0.7166     | 20.32      |
>
> ---
>
> **Q2:** The ~2× speedup is solid but not as high as some recent approaches (e.g., TaylorSeer ~5×).
>
> **A2:**  Thanks for the comment. **As shown in Table R2.1, at similar speedup levels (~5×), MagCache can achieve a superior performance than TaylorSeer on HunyuanVideo with significantly less extra memory cost.** **MagCache with ~2x speedup targets a balanced trade-off between speed and quality, allowing users to enjoy acceleration without sacrificing lots of quality.** If users desire higher speedups, MagCache can be configured with larger $K$ and $\delta$ values to further accelerate inference.
>
> ---
>
> **Q3:** The experiments are conducted on two large text-to-video diffusion models (Open-Sora and Wan), but broader validation is somewhat limited. The method’s generality would be better established by testing on more diverse setups (e.g., CogVideoX).
>
> **A3:** Thanks for valuable suggestions. In the Table 5 of supplementary material, we have included a comparison on HunyuanVideo (540P) and Flux. To strengthen the generality of our approach, we futher expand our evaluation to include HunyuanVideo (480P & 540P), CogVideoX, and Flux. **As shown in Table R2.2, MagCache consistently delivers strong performance across all various models and resolutions, demonstrating its robustness and generality.**
>
> Moreover, **we have also qualitatively validated MagCache on additional models such as FramePack, VACE, Flux Kontext, Chroma, and OmniGen2**, and will release demos and code in our open-source repository.
>
>
>
> **Table R2.2 Generalization of MagCache across Diverse Models and Resolutions.**
>
> | **Method**                           | **Speedup ↑** | **Latency (s) ↓** | **LPIPS ↓** | **SSIM ↑** | **PSNR ↑** |
> | ------------------------------------ | ------------- | ----------------- | ----------- | ---------- | ---------- |
> | **Hunyuan Video** (65 frames, 480P)  |               |                   |             |            |            |
> | Hunyuan Video (T=50)                 | 1×            | 177.50            | -           | -          | -          |
> | DuCa                                 | 4.62×         | 38.41             | 0.3516      | 0.6443     | 16.16      |
> | TeaCache-slow                        | 2.97×         | 59.76             | 0.1759      | 0.7927     | 20.23      |
> | TeaCache-fast                        | 4.55×         | 39.01             | 0.4377      | 0.6216     | 16.07      |
> | TaylorSeer (N=3, O=1)                | 3.00×         | 59.16             | 0.1353      | 0.8335     | 21.42      |
> | TaylorSeer (N=5, O=1)                | **5.00×**     | 35.50             | 0.2691      | 0.7039     | 16.79      |
> | MagCache-slow                        | 3.10×         | 57.20             | **0.1283**  | **0.8372** | **24.38**  |
> | MagCache-fast                        | **5.00×**     | **35.45**         | 0.2656      | 0.7166     | 20.32      |
> | **HunyuanVideo** (129 frames, 540P)  |               |                   |             |            |            |
> | HunyuanVideo (T = 50)                | 1×            | 1163              | -           | -          | -          |
> | TeaCache-slow                        | 1.63×         | 712               | 0.1832      | 0.7876     | 23.87      |
> | TeaCache-fast                        | 2.26×         | 514               | 0.1971      | 0.7744     | 23.38      |
> | MagCache-slow                        | 2.25×         | 516               | **0.0377**  | **0.9459** | **34.51**  |
> | MagCache-fast                        | **2.63×**     | **441**           | 0.0626      | 0.9206     | 31.77      |
> | **CogVideoX 2B** (49 frames, 480P)   |               |                   |             |            |            |
> | CogVideoX (T = 50)                   | 1×            | 74.10             | -           | -          | -          |
> | TeaCache                             | 2.30×         | 32.20             | 0.1221      | 0.8815     | 27.08      |
> | MagCache                             | **2.37×**     | **31.15**         | **0.0787**  | **0.9210** | **30.44**  |
> | **Flux** (Text-to-Image 1024 × 1024) |               |                   |             |            |            |
> | Flux (T = 28)                        | 1×            | 14.26             | -           | -          | -          |
> | TeaCache-slow                        | 2.00×         | 7.11              | 0.2687      | 0.7746     | 20.14      |
> | TeaCache-fast                        | 2.52×         | 5.65              | 0.3456      | 0.7021     | 18.17      |
> | MagCache-slow                        | 2.57×         | 5.53              | **0.2043**  | **0.8883** | **24.46**  |
> | MagCache-fast                        | **2.82×**     | **5.05**          | 0.2635      | 0.8093     | 21.35      |

---

> > ### Author Response · Authors · 2025-08-05
> >
> > Thank you for your valuable time in reviewing our paper. We would greatly appreciate it if you could let us know whether our response has addressed your concerns. If there are any remaining questions, we would be sincerely grateful for the opportunity to further clarify and will make every effort to respond.

---

> ### Author Response · Authors · 2025-08-06
>
> Dear Reviewer AVyF,
>
> Thank you for your valuable time in reviewing our manuscript. As the discussion phase is ending soon,  we would greatly appreciate it if you could consider our response and supplementary experiments in your final assessment. MagCache outperforms both TaylorSeer and DuCa in quality, speed, and memory efficiency (Table R2.1), and we validate its effectiveness on more video/image generation models, including CogVideoX, HunyuanVideo, and Flux (Table R2.2). We hope that our response and the supplementary experiments have satisfactorily addressed your concerns.
>
> Thank you for your valuable time!
>
> Sincerely,
>
> Authors of Paper 16663

---

> ### Author Response · Authors · 2025-08-08
>
> Dear Reviewer AVyF,​
>
> I hope this message finds you well. Please accept my sincere apologies for following up again; we understand your time is extremely valuable. However, I’m writing with urgency as tomorrow marks the final day of our paper’s rebuttal review period, and we haven’t yet received your feedback on our rebuttal.​
>
> We’ve worked diligently to address all concerns from your initial review, hoping our rebuttal clarifies our work. With only one day left until the deadline, we’d be deeply grateful if you could spare a few moments to review it at your convenience. Your insights are critical to our paper’s evaluation at this stage, and we hope our rebuttal has adequately addressed the points you raised before the deadline closes.​
>
> Thank you again for your time and consideration. We apologize for any inconvenience and look forward to your input.​
>
> Sincerely,
>
> Authors of Paper 16663

---

### Official Review · Reviewer_vpsp · 2025-07-02

**Clarity:** 2
**Significance:** 3
**Originality:** 2
**Rating:** 4
**Confidence:** 3

**Summary:**

This paper introduces MagCache, a magnitude-based caching method to accelerate the inference of video diffusion models, after analyzing the relationships between magnitude residuals across diffusion timesteps. The authors propose accurate error modeling and adaptive caching strategy to adaptively skip consecutive diffusion timesteps until its accumulated error exceeds the predefined threshold or maximum skip length. Experiments demonstrate the over 2x acceleration of MagCache on Open-Sora and Wan 2.1.

**Questions:**

1. How sensitive is the proposed magnitude-aware caching method to the choice of the single calibration sample? Would the choice affect the generation quality? While Figure 1 shows consistency across three prompts, would an outlier or particularly complex prompt result in a less representative curve?

2. Are the hyperparameters dependent on the denoising scheduler? Should these hyperparameters be recalibrated when altering the denoising scheduler or the number of denoising steps?

**Ethical Concerns:**

["NO or VERY MINOR ethics concerns only"]

**Final Justification:**

I would like to maintain my original positive score for this work.

**Limitations:**

Yes.

**Paper Formatting Concerns:**

No.

**Quality:**

3

**Strengths And Weaknesses:**

Strengths
The analysis in Figure 1 clearly demonstrates the motivation of this paper to use magnitude residual as the calibration information.
The proposed method is a plug-and-play solution requires only a random sample for calibration.
The experimental results demonstrate the acceleration (over 2x) of video diffusion models’ inference.

Weaknesses
The method's performance relies on the careful tuning of two key hyperparameters $\delta$ and $K$ for each model.
There are step reduction methods for accelerating diffusion models, which may have greater variations between steps. MagCache’s may not be useful for few-step diffusion models.
Comparison with other relevant caching-based video diffusion acceleration methods is missing, such as AdaCache and FasterCache.

---

> ### Author Rebuttal · Authors · 2025-07-30
>
> **Q1:** The performance relies on the tuning of two key hyperparameters $\delta$ and $K$ for each model.
>
> **A1:** Thanks for the question. The varying $\delta$ and $K$ settings across models were chosen to ensure fair comparisons at similar speedups. In practice, **MagCache performs robustly with default parameters across different models**, requiring only minor $\delta$ adjustments. We offer two predefined modes:
>
> - **Slow mode: $K = 2$, $\delta = 0.06$ (default)**
> - **Fast mode: $K = 4$, $\delta = 0.06$ (default)**
>
> Here, $K$ controls speedup mode, and $\delta$ adjusts the quality–speed trade-off within the selected mode. In Table R1.1, varying $\delta$ offers fine control with just 1–2 adjustments. **We recommend the default slow mode for general use, and our ComfyUI interface makes parameter tuning fast and user-friendly.**
>
> **Table R1.1 Parameter Configurations in Slow and Fast inference mode.**
>
> | **Mode**                     | $K$   | $\delta$ | **Speedup ↑** | **SSIM ↑** | **PSNR ↑** |
> | :--------------------------- | ----- | -------- | ------------- | ---------- | ---------- |
> | **MagCache-slow (Wan2.1)**   | **2** | **0.06** | 2.0           | 0.838      | 24.5       |
> |                              |       | 0.12     | 2.1           | 0.827     | 24.3       |
> |                              |       | 0.03     | 1.9           | 0.842      | 24.6       |
> | **MagCache-fast (Wan2.1)**   | **4** | **0.06** | 2.4           | 0.774      | 22.3       |
> |                              |       | 0.12     | 2.7           | 0.757      | 22.2       |
> |                              |       | 0.03     | 2.0           | 0.782      | 22.5       |
> | **MagCache-slow (OpenSora)** | **2** | **0.06** | 1.8           | 0.814      | 23.9       |
> |                              |       | 0.12     | 1.9           | 0.813      | 23.8       |
> |                              |       | 0.03     | 1.7           | 0.816      | 24.0       |
> | **MagCache-fast (OpenSora)** | **4** | **0.06** | 2.1           | 0.763      | 21.9       |
> |                              |       | 0.12     | 2.4           | 0.754      | 21.7       |
> |                              |       | 0.03     | 2.0           | 0.766      | 22.0       |
>
> ---
>
> **Q2:** MagCache’s may not be useful for few-step diffusion models.
>
> **A2:** Despite larger residual variations in few-step models, **MagCache still provides acceleration by reusing residuals with small differences**. On FusionX (a 10-step distilled Wan2.1 14B), MagCache achieves a 1.66× speedup with minimal quality loss(Table R1.2). In contrast, directly reducing the steps to 6 significantly degrades quality, highlighting the advantage of MagCache.
>
> **Table R1.2: Evaluation of MagCache on Few-Step Distilled Model (Wan 2.1 14B FusionX, 33 frames, 480P)**
>
> | **Method**                    | **Speedup ↑** | **SSIM ↑** | **PSNR ↑** |
> | ----------------------------- | ------------- | ---------- | ---------- |
> | FusionX (T = 10)              | 1             | -          | -          |
> | FusionX (T = 6)               | 1.66          | 0.647      | 20.3       |
> | MagCache (T=10, skip 4 steps) | 1.66          | **0.786**  | **24.2**   |
>
> ---
>
> **Q3:** Comparison with other caching-based acceleration methods, such as AdaCache and FasterCache.
>
> **A3:** Thanks for the valuable suggestion. We compare MagCache with AdaCache, FasterCache, DuCa, and TaylorSeer across Open-Sora, Wan2.1, and Hunyuan in Table R1.3. On Open-Sora, MagCache matches AdaCache’s speedup but outperforms it in quality with significantly less extra memory (0.1G vs 11.3G). It also surpasses FasterCache in performance, speed, and memory efficiency **(0.1 G vs. 11.3 G)** on Open-Sora. **This is because MagCache caches only a single residual instead of layer-wise outputs like AdaCache and FasterCache.** Additionally, MagCache consistently achieves superior performance on both the Wan and Hunyuan models. **Across all models, MagCache offers a strong balance of speed, quality, and memory usage.**
>
>
>
> **Table R1.3: Comparison with Other Cache-based Methods across Various Models.**
>
> | **Method**                                  | Speedup ↑ | Extra Memory↓ | SSIM ↑    | PSNR ↑   |
> | ------------------------------------------- | --------- | ------------- | --------- | -------- |
> | **Open-Sora 1.2 (T = 30, 51 frames, 480P)** | 1         | -             | -         | -        |
> | FasterCache                                 | 1.7       | 11.3G         | 0.825     | 23.2     |
> | AdaCache                                    | 2.1       | 11.3G         | 0.786     | 21.0     |
> | DuCa                                        | 2.0       | 23.9G         | 0.765     | 19.9     |
> | TeaCache-slow                               | 1.4       | **0.1G**      | 0.840     | 23.6     |
> | TeaCache-fast                               | 2.0       | **0.1G**      | 0.743     | 18.9     |
> | MagCache-slow                               | 1.4       | **0.1G**      | **0.885** | **26.9** |
> | MagCache-fast                               | **2.1**   | **0.1G**      | 0.826     | 23.3     |
> | **Wan 2.1 (T = 50, 81 frames, 480P)**       | 1         |               | -         | -        |
> | TeaCache                                    | 2.1       | **0.5G**      | 0.657     | 18.1     |
> | TaylorSeer(N=2, O=1)                        | 2.0       | 40G           | 0.522     | 15.0     |
> | MagCache-fast                               | **2.6**   | **0.5G**      | **0.749** | **21.5** |
> | **Hunyuan Video (T=50, 65 frames, 480P)**   | 1         |               | -         | -        |
> | AdaCache                                    | 2.6       | -             | 0.695     | 17.0     |
> | TeaCache                                    | 2.9       | 0.1G          | 0.792     | 20.2     |
> | TaylorSeer (N=3, O=1)                       | 3.0       | 46.2G         | 0.833     | 21.4     |
> | MagCache                                    | **3.1**   | **0.1G**      | **0.837** | **24.3** |
>
> ---
>
> **Q4:** Sensitivity to the calibration sample. Does prompt selection impact the quality? Would an outlier or particularly complex prompt result in a less representative curve?
>
> **A4: MagCache is robust to the selection of the calibration sample**, and generation quality remains consistent even with outliers. We first evaluate 944 prompts from VBench and observe very low variance in magnitude ratios and cosine distances(Table R1.4). The magnitude ratios exhibit a stable, monotonically decreasing trend, with sharper drops in the final 20% steps. The cosine distance trends are also stable across prompts.
>
> We also test MagCache with three calibration prompt: a random prompt, all 944 prompts, and an outlier prompt (farthest from the average curve). **All three yield similar speedup and visual quality (Table R1.5)**, confirming MagCache's robustness.
>
>
>
> **Table R1.4 Statistics of Magnitude Ratios and Cosine Distance across 944 Vbench prompts.** `0.9965 (± 0.0004) denotes mean ± standard deviation`
>
> | Diffusion Process (%) | Magnitude Ratio (Wan 2.1) | Cosine Distance  (Wan 2.1) |
> | --------------------- | ------------------------- | -------------------------- |
> | 0–20                  | 0.9965 (±0.0004)          | 0.0008 (±0.0003)           |
> | 20–40                 | 0.9951 (±0.0002)          | 0.0003 (±0.0003)           |
> | 40–60                 | 0.9928 (±0.0002)          | 0.0001 (±0.0001)           |
> | 60–80                 | 0.9851 (±0.0002)          | 0.0002 (±0.0001)           |
> | 80–100                | 0.9285 (±0.0019)          | 0.0054 (±0.0009)           |
>
> **Table R1.5 Influence of Calibration Prompt.**
>
> | Calibration Prompt    | Speedup ↑ | SSIM ↑ | PSNR ↑ |
> | --------------------- | --------- | ------ | ------ |
> | Random Prompt (Ours)  | 2.1       | 0.813  | 23.4   |
> | 944 Prompts (Average) | 2.1       | 0.816  | 23.5   |
> | Outlier Prompt        | 2.1       | 0.810  | 23.3   |
>
> ---
>
> **Q5:** Are the hyperparameters dependent on the denoising scheduler? Should hyperparameters be recalibrated when altering denoising scheduler or number of denoising steps?
>
> **A5: MagCache’s hyperparameters are robust across different schedulers and numbers of steps. As shown in Table R1.6 and Table R1.7, magnitude ratios calibrated with one scheduler or step setting can generalize well to others without recalibration.** For instance, magnitude ratios calibrated under UniPC can be applied to inference using DPM++, and vice versa, with minimal impact on quality or speedup. When steps change, we use nearest-neighbor interpolation to align the ratio curve. Note that using the same inference scheduler or number of steps tends to yield consistent results.
>
>
>
> **Table R1.6 Robustness of Magnitude Ratios to Different Schedulers.**
>
> | Calibrated Scheduler | Inference Scheduler | Speedup ↑ | SSIM ↑ | PSNR ↑ |
> | -------------------- | ------------------- | --------- | ------ | ------ |
> | UniPC                | UniPC               | 2.1       | 0.827  | 24.2   |
> | UniPC                | DPM++               | 2.1       | 0.841  | 24.3   |
> | DPM++                | DPM++               | 2.1       | 0.841  | 24.3   |
> | DPM++                | UniPC               | 2.1       | 0.827  | 24.2   |
> | Euler                | Euler               | 2.1       | 0.85   | 25.1   |
> | Euler                | UniPC               | 2.1       | 0.827  | 24.2   |
> | UniPC                | Euler               | 2.1       | 0.85   | 25.1   |
>
> **Table R1.7 Robustness of Magnitude Ratios to Different Numbers of Steps.**
>
> | Calibrated Steps | Inference Steps | Speedup ↑ | SSIM ↑ | PSNR ↑ |
> | ---------------- | --------------- | --------- | ------ | ------ |
> | 50               | 50              | 2.1       | 0.827  | 24.3   |
> | 30               | 50              | 2.1       | 0.827  | 24.3   |
> | 50               | 30              | 2.1       | 0.711  | 21.0   |
> | 30               | 30              | 2.1       | 0.711  | 21.0   |

---

> > ### Comment · Reviewer_vpsp · 2025-08-05
> >
> > Thanks for your efforts in the detailed rebuttal.
> >
> > My concerns has been taken good care of. I would like to keep my positive score for this work.

---

> ### Author Response · Authors · 2025-08-08
>
> Dear Reviewer vpsp,
>
> Thank you sincerely for your valuable time to review our paper and providing such insightful feedback. We deeply appreciate the thought and effort you have invested in this process. We wanted to update you that the **reviewer AVyF and 9jpp , who initially assigned scores of 3, have not yet joined any discussion.** Given this, we would be extremely grateful if you might be willing to reconsider our work with a higher score.
>
> **Any slight adjustment to your score, or your continued support when engaging in discussions with the Area Chair, would mean a great deal to us as we strive to address feedback and strengthen our submission.**
>
> Thank you again for your support and consideration.
>
> Sincerely,
>
> Authors of Paper 16663

---

### Author Response · Authors · 2025-08-04
**Gentle Reminder**

Dear All Reviewers,

We hope our response and supplementary experiments have successfully addressed your concerns. As the discussion period nears its end, we would be grateful if you would consider our clarifications and new results in your final assessment. We are looking forward to your further comments or responses.

Thank you for your valuable time!

Sincerely,

Authors of Paper #16663

---

### Note · Authors · 2025-08-12

**Dear Area Chair,**

We thank all reviewers and ACs for their time and effort in evaluating our work.

Our paper introduces **MagCache**, a novel caching strategy to accelerate video generation in diffusion models. MagCache leverages a newly discovered *magnitude law*, where the magnitude ratio of successive residual outputs decreases monotonically across different models and prompts. By combining an error modeling mechanism with an adaptive caching strategy, MagCache selectively skips unimportant timesteps, achieving substantial inference ***speedups (2.1×–3.1×) for OpenSora, Wan 2.1, Hunyuan Video, CogVideoX, and Flux***, while preserving high visual fidelity.

We appreciate the positive feedback from reviewer vpsp and YTog who engaged deeply with our work. **Reviewer vpsp** recognized ***our motivation*** for using the magnitude ratio as calibration information and noted that MagCache is a plug-and-play solution delivering over 2× speedup with only a single random sample for calibration. **Reviewer YTog** acknowledged our novel unified magnitude decay law and its role in providing a ***theoretical basis*** for timestep skipping. YTog also highlighted the significant acceleration, superior visual quality, and negligible calibration cost compared to existing methods. Both reviewers agreed that our rebuttal satisfactorily addressed their concerns.

**However, reviewer AVyF and 9jpp, who initially gave negative scores, did not participate in any rebuttal discussion**, which may hinder a fair reassessment. For **reviewer AVyF**, we presented evidence that MagCache outperforms recent methods in quality, speed, and memory efficiency (Table R2.1), and validated its effectiveness on additional video generation models (Table R2.2). For **reviewer 9jpp**, we demonstrated MagCache’s robustness and generality across more models and prompts (Tables R3.1–R3.5), and we included a theoretical analysis of Equation (5) in A2.

We believe the clarification and supplementary experiments comprehensively address the concerns raised by AVyF and 9jpp. **We trust that the Area Chair will consider these points in making a fair assessment, particularly in light of the fact that reviewer AVyF and 9jpp did not engage in the rebuttal and their initial evaluations may be based on a partial understanding.**

Thank you for your time and consideration.

Best regards,

Authors of Paper 16663

---

### Decision · Program_Chairs · 2025-09-17

**Decision:**

Accept (poster)

**Comment:**

The paper was favorably reviewed by 4 experts, that recommended acceptance. The paper proposes a testing time speed up method for video diffusion models, that takes into consideration the magnitude residuals across time steps. The method gets 2x acceleration on most recent models, compared to prior works in this area (TeaCache). The AC agrees and recommends acceptance. Congrats!